# ReMoDetect: Reward Models Recognize Aligned LLM's Generations

**Hyunseok Lee**[*1]**, Jihoon Tack**[*,1]**, Jinwoo Shin**[1]
[1]Korea Advanced Institute of Science and Technology
{hs.lee,jihoontack,jinwoos}@kaist.ac.kr

## Abstract

The remarkable capabilities and easy accessibility of large language models (LLMs) have significantly increased societal risks (e.g., fake news generation), necessitating the development of LLM-generated text (LGT) detection methods for safe usage. However, detecting LGTs is challenging due to the vast number of LLMs, making it impractical to account for each LLM individually; hence, it is crucial to identify the common characteristics shared by these models. In this paper, we draw attention to a common feature of recent powerful LLMs, namely the alignment training, i.e., training LLMs to generate human-preferable texts. Our key finding is that as these *aligned LLMs* are trained to maximize the human preferences, they generate texts with higher estimated preferences even than human-written texts; thus, such texts are easily detected by using the *reward model* (i.e., an LLM trained to model human preference distribution). Based on this finding, we propose two training schemes to further improve the detection ability of the reward model, namely (i) continual preference fine-tuning to make the reward model prefer aligned LGTs even further and (ii) reward modeling of Human/LLM mixed texts (a rephrased texts from human-written texts using aligned LLMs), which serves as a median preference text corpus between LGTs and human-written texts to learn the decision boundary better. We provide an extensive evaluation by considering six text domains across twelve aligned LLMs, where our method demonstrates state-of-the-art results. Code is available at https://github.com/hyunseoklee-ai/ReMoDetect.

## 1  Introduction

Large Language models (LLMs) [8, 41] have significantly accelerated progress in natural language processing (NLP) and thus become a core technology in various real-world applications used by millions of users, such as coding assistants [9], search engines [46], and personal AI assistants [12]. However, due to their remarkable capabilities, they also lead to multiple misuses, which raises serious safety concerns, e.g., fake news generation [32], plagiarism [22], and malicious comments [23] using LLMs. In this regard, developing automatic LLM-generated text (LGT) detection frameworks is becoming more crucial for the safe usage of LLMs [32, 11, 13].

To tackle this issue, there have been several efforts to build LGT detectors [21, 2]. Here, one line of the literature proposes to train a binary classifier using the human-written texts and LGTs [20, 6]. However, assuming specific knowledge (e.g., training with LGTs from specific LLMs) may introduce a bias to the detector, thus requiring a careful training. In this regard, another line of work focuses on zero-shot detection (i.e., detecting with a frozen LLM), aiming to capture a useful common characteristic of LLMs for effective detection [20, 34]. Despite their significant efforts, it is still quite challenging (and had relatively less interest) to detect texts generated by recent powerful LLMs such as GPT-4 [26] and Claude [5], which is a realistic and important LGT detection scenario [11, 13].

---

[*]Equal contribution

38th Conference on Neural Information Processing Systems (NeurIPS 2024).

Table 1: AUROC (%) of LLM-generated text detection methods on WritingPrompts from the Fast-DetectGPT benchmark, where GPT4 is used for text generation. 'Reward model' indicates the detection using the reward score of the pre-trained reward model. The bold denotes the best result.

| Method | AUROC |
|---|---|
| Log-likelihood [20] | 85.5 |
| DetectGPT [11] | 80.9 |
| Fast-DetectGPT [13] | 96.1 |
| Reward model | 92.8 |
| **ReMoDetect** | **98.8** |

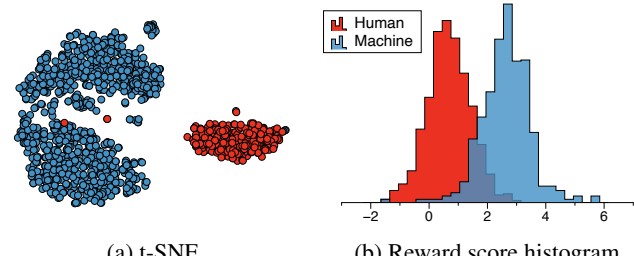

(a) t-SNE      (b) Reward score histogram

Figure 1: **Motivation**: Aligned LGTs and human-written texts are easily distinguishable by using the reward model. We visualize the (a) t-SNE of the reward model's final feature and the (b) histogram of the predicted reward score. Here, 'Machine' indicates the text generated by GPT3.5/GPT4 Turbo, Llama3-70B, and Claude on the Reuters domain.

In this regard, we draw attention to a common yet important feature of recent powerful LLMs: the *alignment training* [27, 30, 19], i.e., training LLMs to generate human-preferable texts. For instance, one way to align LLMs is to (i) train a *reward model* that reflects the human preference distribution and (ii) then fine-tune the LLM to maximize the predicted reward of the generated text.

**Contribution.** In this paper, we present a somewhat interesting observation by using the reward model: as aligned LLMs are optimized to maximize human preferences, they generate texts with higher predicted rewards even compared to human-written texts (see Figure 1).[2] Based on this, one can easily distinguish LLM-generated texts from human-written texts by simply using the predicted score of the reward model as the detection criteria, e.g., AUROC of 92.8% when detecting GPT4 generated texts (in Table 1). Inspired by this, we suggest further exploiting the reward model for aligned LGT detection by enhancing the score separation between the human- and LGTs.

We propose ReMoDetect, a novel and effective aligned LGT detection framework using the reward model. In a nutshell, ReMoDetect is comprised of two training components to improve the detection ability of the reward model. First, to further increase the separation of the predicted reward between LGTs and human-written texts, we continually fine-tune the reward model to predict even higher reward scores for LGTs compared to human-written-texts while preventing the overfitting bias using the replay technique [31]. Second, we generate an additional preference dataset for reward model fine-tuning, namely the Human/LLM mixed text; we partially rephrase the human-written text using LLM. Here, such texts are used as a median preference corpus among the human-written text and LGT corpora, enabling the detector to learn a better decision boundary.

We demonstrate the efficacy of ReMoDetect through extensive evaluations on multiple domains and aligned LLMs. Overall, our experimental results show strong results of ReMoDetect where it significantly outperforms the prior detection methods, achieving state-of-the-art performance. For instance, measured with the average AUROC (%) across three text domains in Fast-DetectGPT benchmark [13], ReMoDetect demonstrates superior performance over the prior work from 90.6→97.9 on the GPT-4 and 92.6→98.6 on Claude3 Opus generated texts. Moreover, we highlight that ReMoDetect is robust in multiple aspects, including robustness against rephrasing attacks (i.e., detecting rephrased text originating from LGTs), detection text length, and unseen distributions.

## 2 Related Work

**Large Language Model (LLM) generated text detection.** There are several approaches to detecting text generated by LLMs, mainly categorized in two: (i) training supervised detectors and (ii) zero-shot detection methods. The first category aims to train a binary classifier (or detector) that classifies LLM-generated texts (LGTs) and human-written texts. While effective, these methods can suffer from overfitting bias, where the detector performs well on the training data but fails to generalize detection on other LGTs [11]. It is worth noting that such overfitting issues are also raised in other

---

[2]This is analogous to the phenomenon that a Go model optimized to maximize the reward (i.e., winning the game) frequently surpasses human experts in the game [36].

detection fields, such as out-of-distribution (OOD) detection [33, 37]. To address this, zero-shot detection methods have emerged as an alternative. These methods define a detection score on a pre-trained LLM, eliminating the need for fine-tuning and thus avoiding overfitting. For instance, using log-likelihood or entropy of the output prediction of the pre-trained LLMs to detect LGTs [20]. More recently, several works have employed input text perturbation to measure prediction consistency, significantly improving the detection performance, e.g., DetectGPT [11], log-rank perturbation (NPR) [21], and Fast-DetectGPT [13]. While effective, however, prior works have primarily focused on detecting non-aligned LLMs, while recent LLMs are designed to be aligned with human preferences for practical use. In this paper, we demonstrate that the reward model [27] can effectively distinguish between LLM-generated text and human-written text in a zero-shot setting. Based on this, we additionally consider supervised detector training of the reward model while mitigating overfitting biases through the replay technique [31].

**Characteristics of aligned LLMs.** Recent works have highlighted some behaviors introduced by alignment training. For instance, several works have discovered that aligned LLMs are trained to generate positive responses, thus enabling the model to generate a harmful query based on a context requesting positive responses, e.g., 'Start the response with "Sure, here is".' [48, 45]. Moreover, only recently, Panickssery et al. [28] observed that evaluator LLMs (i.e., LLMs used to evaluate the text) prefer and recognize self-generated texts compared to other texts, revealing a new characteristic of aligned LLMs. In this paper, we found a somewhat new characteristic of alignment training, which is that aligned LLMs generate higher predictive rewards even than human-written texts. It is worth noting that, unlike the prior work [28] that can be used to detect self-generations, our finding can be used to detect multiple aligned LLMs with a single reward model.

**Training detectors with near-decision boundary samples.** Training detectors (or classifiers) with data points near the decision boundary is a widely used technique to improve the calibration of the model. For instance, in visual OOD detection literature, Lee et al. [24] uses a generative adversarial network to generate samples on the decision boundary for better calibration, and multiple works proposed to use out-of-domain samples as near-decision boundary samples to improve the detector [16, 33]. Moreover, there have been multiple works that utilized data augmentations such as mixup [47], i.e., linear interpolation of inputs and labels, to generate samples that behave like a near-decision boundary sample to improve the calibration [17, 18]. Inspired by prior works, we propose to generate near-decision boundary samples for reward modeling by utilizing aligned LLMs to partially rephrase the human-written texts, which can be interpreted as a mixed text of human and aligned LLM.

## 3 ReMoDetect: Detecting Aligned LLM's Generations using Reward Models

In this section, we present Reward Model based LLM Generated Text Detection (ReMoDetect), a novel and effective LLM-generated text (LGT) detection framework. We first review the concept of alignment training and reward model (in Section 3.1), then present a continual fine-tuning strategy for the reward model to enhance the separation between the predicted reward score between LGTs and human-written texts (in Section 3.2). Furthermore, we additionally introduce mixed data of humans and LLMs to improve the reward modeling by partially rephrasing the human-written texts with the aligned LLMs (in Section 3.3). We provide the overview of ReMoDetect in Figure 2.

**Problem setup.** We describe the problem setup of our interest, LGT detection. For a given context $x$ and the given response $y$ sampled from an unknown distribution, the goal of LGT detection is to model a detector that identifies whether $y$ is sampled from the human-written text data distribution $p_{\text{data}}(y|x)$ or from a large language model (LLM; $\mathcal{M}$), i.e., $\mathcal{M}(y|x)$. To this end, existing methods for LGT detection define a score function upon the detector model that a high value heuristically represents that $y$ is from the human-written text data distribution.

### 3.1 Alignment Training and Reward Modeling

Recent LLMs are trained in two sequential steps: (i) unsupervised pre-training on a large text corpus [1, 8] then (ii) training LLMs to generate texts that align with human preferences (also known as alignment training) [27, 30, 19]. In this paper, we found that this alignment training can force the LLM to generate texts that are too close to human preferences, even compared to human-written texts. To quantify such a value of the given text, we use the prediction of the reward model [27], which is trained to reflect human preferences.

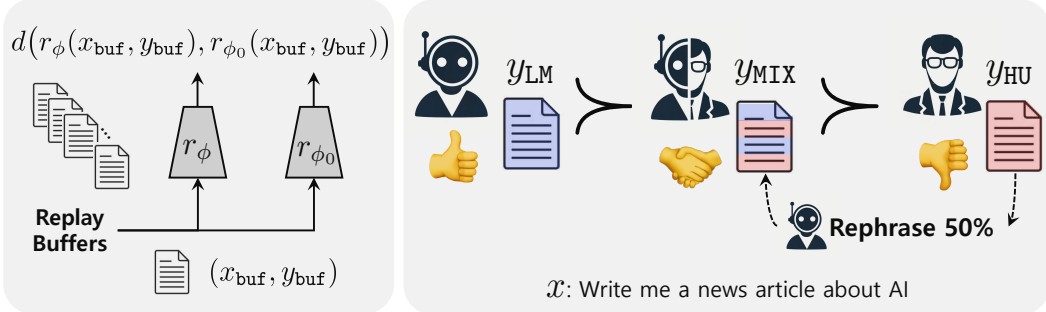

**Continual Preference Tuning**      **Reward Modeling with Mixed Responses**

Figure 2: Overview of Reward Model based LLM Generated Text Detection (ReMoDetect): We continually fine-tune the reward model $r_\phi$ to prefer aligned LLM-generated responses $y_{\text{LM}}$ even further while preventing the overfitting by using the replay technique: $(x_{\text{buf}}, y_{\text{buf}})$ is the replay buffer and $r_{\phi_0}$ is the initial reward model. Moreover, we generate a human/LLM mixed text $y_{\text{MIX}}$ by partially rephrasing the human response $y_{\text{HU}}$ using the aligned LLM, which serves as a median preference data compared to $y_{\text{LM}}$ and $y_{\text{HU}}$, i.e., $y_{\text{LM}} \succ y_{\text{MIX}} \succ y_{\text{HU}} \mid x$, to improve the reward model's detection ability.

**Reward model.** For a given context $x$ and the corresponding response $y$, the reward model $r_\phi(x, y) \in \mathbb{R}$ parameterized by $\phi$, models the human preference of $(x, y)$. To train such a model, one of the most conventional ways is to use the Bradley-Terry model [7] based on the collection of preference labels: the labeler is required to choose the better response among two responses based on the given context $x$, formally as $y_w \succ y_l \mid x$ where $y_w$ and $y_l$ indicates the preferred and dispreferred response, respectively. Then the Bradley-Terry model defines the human preference distribution as follows:

$$p(y_w \succ y_l \mid x) = \frac{\exp\left(r_\phi(x, y_w)\right)}{\exp\left(r_\phi(x, y_w)\right) + \exp\left(r_\phi(x, y_l)\right)}.$$

By considering the reward modeling as a binary classification problem, one can minimize the following negative log-likelihood loss to train the reward model:

$$\mathcal{L}_{\text{RM}}(x, y_w, y_l) := -\log \sigma(r_\phi(x, y_w) - r_\phi(x, y_l)).$$

where $\sigma(\cdot)$ is the logistic function.

**Motivation.** By utilizing the pre-trained reward model, we observed that the predicted reward score of aligned LGT is higher than the human-written text (in Figure 1 and more examples are presented in Section 4.2). This indicates that the alignment training optimizes the LLM to generate texts with high human preferences, which makes the LLM generate texts that are actually far away from the human-written text data distribution $p_{\text{data}}(y|x)$. Inspired by this observation, we suggest utilizing the reward model for aligned LGT detection.

### 3.2 Continual Preference Tuning: Increasing the Separation Gap of the Predicted Reward

Based on our observation, we suggest further increasing the separation gap of the predicted rewards between aligned LGTs and human-written texts. To this end, we use the Bradley-Terry model to continually fine-tune the reward model so that the model prefers LGTs even further compared to human-written texts. Furthermore, it is important to consider the overfitting issue when fine-tuning the reward model as assuming specific prior knowledge may introduce a bias to the detector [37, 11, 13], e.g., training detector with LGTs of some specific LLMs may not generalize detection on other LLM's generated texts. In this regard, we prevent overfitting by regularizing the prediction change of the current reward model from the initial reward model using replay buffers [31], i.e., samples used for training the initial reward model. Formally, for a given human-written text/LGT pair $(y_{\text{HU}}, y_{\text{LM}})$ based on the context $x$, and the reward model's parameter $\phi$, the training objective is as follows:

$$\mathcal{L}_{\text{cont}} := \mathcal{L}_{\text{RM}}(x, y_{\text{LM}}, y_{\text{HU}}) + \lambda \, d\big(r_\phi(x_{\text{buf}}, y_{\text{buf}}), r_{\phi_0}(x_{\text{buf}}, y_{\text{buf}})\big), \tag{1}$$

where $\phi_0$ is the pre-trained reward model's parameter, $\lambda$ is a parameter for controlling the deviation from the initial reward model, $d(\cdot, \cdot)$ is the $\ell_2$ distance function, and $(x_{\text{buf}}, y_{\text{buf}})$ is the replay buffer.

### 3.3 Reward Modeling of Human and LLM Mixed Dataset

We suggest utilizing the human and LLM mixed dataset to further improve the detection performance. Specifically, we partially rephrase human-written texts using aligned LLMs to generate the mixed dataset, which are considered as median preference datasets between LGTs and human-written texts. Note that such a technique introduces new samples that behave like a reasonable near-decision boundary sample, which enables the detector to learn a better decision boundary. For instance, multiple out-of-distribution detection methods utilize generated samples [24] such as mixup data [47, 17] as a near-decision boundary sample to improve the detector's calibration.

Concretely, for a given context $x$ and the human-written response $y_{\text{HU}}$, we partially rephrase the response with a ratio of $p$, using LLM $\mathcal{M}_{\text{rep}}$, i.e., $y_{\text{MIX}} := \mathcal{M}_{\text{rep}}(y_{\text{HU}}|x, p)$. We consider $y_{\text{MIX}}$ as a median preference response between human-written text $y_{\text{HU}}$ and LGT $y_{\text{LM}}$ which is formally described as: $y_{\text{LM}} \succ y_{\text{MIX}} \succ y_{\text{HU}} \mid x$. Since the Bradely-Terry modeling assumes binary classification, we consider dividing the triplet into three binary classification problems, i.e., $y_{\text{LM}} \succ y_{\text{HU}} \mid x$, $y_{\text{LM}} \succ y_{\text{MIX}} \mid x$, and $y_{\text{MIX}} \succ y_{\text{HU}} \mid x$. Therefore, the final training objective of ReMoDetect additionally considers the mixed dataset's preference modeling in addition to Eq. (1), which is as follows:

$$\mathcal{L}_{\text{ours}} := \mathcal{L}_{\text{cont}} + \beta_1 \, \mathcal{L}_{\text{RM}}(x, y_{\text{MIX}}, y_{\text{HU}}) + \beta_2 \, \mathcal{L}_{\text{RM}}(x, y_{\text{LM}}, y_{\text{MIX}}) \tag{2}$$

where $\beta_1$ and $\beta_2$ are parameters that chooses the contribution of the mixed data $y_{\text{MIX}}$.

**Detection stage.** After training ReMoDetect, we use the predicted reward score $r_\phi(x, y)$ to determine whether the given text is LGT or human-written texts where a higher score indicates LGT. Unlike recent detection schemes that require multiple forwards (for perturbing the input [11, 13]), ReMoDetect only requires a single forward pass, thus showing inference efficiency (in Section 4.3).

## 4 Experiments

We provide an empirical evaluation of ReMoDetect by investigating the following questions:

- Can ReMoDetect detect texts generated from aligned LLMs? (Table 2 & Table 3)
- Do reward models recognize aligned LLM's generations? (Figure 3 & Figure 4)
- Is ReMoDetect robust to rephrasing attacks and challenging setups? (Table 4 & Table 5 & Figure 6)
- How do/Do the proposed components enhance the detection performance? (Figure 5 & Table 7)

Before answering each question, we outline the experimental protocol (more details in Appendix A).

**Evaluation setup.** We mainly report the area under the receiver operating characteristic curve (AUROC) as a threshold-free evaluation metric (results with other metrics are presented in Appendix B.3). Here, the text is written (or generated) in 6 text domains introduced in Fast-DetectGPT [13] and MGTBench [15], including PubMed [29], XSum [35], Reuters [43], Essay [43], and WritingPrompts [4] (each benchmark consists of different types of WritingPrompts, thus denoting the version in [13] as small-sized). In addition to GPT3.5 Turbo, GPT4, and Claude, which are already provided in the benchmark, we consider more aligned LLMs $\mathcal{M}$, including Llama3 70B instruct [41], Claude3 Opus [5] Gemini pro [38], and GPT4 Turbo [26]. We also consider more aligned LLMs, e.g., models trained with direct preference optimization (DPO) [30], in Table 6 and Appendix B.2.

**Training setup of ReMoDetect.** For the main experiment, we use the reward model from OpenAssistant [3], a 500M-sized LLM for efficient training and inference (we also consider other reward models in Section 4.2). We train ReMoDetect with HC3 dataset by following ChatGPT-Detector [6], which consists of human and ChatGPT responses to the same context. For generating Human/LLM mixed datasets, we use Llama3 70B instruct as $\mathcal{M}_{\text{rep}}$ to rephrase 50% ($p = 0.5$) of human-written texts. Unless otherwise specified, we train a single model for ReMoDetect, which is used across all experiments (i.e., we did not train separate ReMoDetect for individual datasets or aligned LLMs).

**Baselines.** We compare ReMoDetect with multiple detection methods, which fall into three categories. First, we consider zero-shot detectors, including Log-likelihood [20], Rank [20], DetectGPT [11], LRR [21], NPR [21], and Fast-DetectGPT [13] where we use GPT families as the base detector (e.g., GPT-J [44]) by following prior works. For supervised detectors, we consider open-source checkpoints of OpenAI-Detector [20] and ChatGPT-Detector [6], which are trained on GPT2 generated texts and HC3 datasets, respectively. Finally, we consider GPTZero [39], a commercial LLM-generated text (LGT) detection method. We also compare ReMoDetect with more baselines in Appendix B.1.

Table 2: AUROC (%) of multiple LGT detection methods, including log-likelihood (Loglik.) [20], Rank [20], DetectGPT (D-GPT) [11], LRR [21], NPR [21], Fast-DetectGPT (FD-GPT) [13], OpenAI-Detector (Open-D) [20], ChatGPT-Detector (Chat-D) [6], and ReMoDetect (Ours). We consider two major LGT detection benchmarks from (a) Fast-DetectGPT [13] and (b) MGTBench [15]. The bold indicates the best result within the group.

(a) Fast-DetectGPT benchmark [13]: PubMed, XSum, and WritingPrompts-small (WP-s)

| Model | Domain | Loglik. | Rank | D-GPT | LRR | NPR | FD-GPT | Open-D | Chat-D | Ours |
|---|---|---|---|---|---|---|---|---|---|---|
| GPT3.5 Turbo | PubMed | 87.8 | 59.8 | 74.4 | 74.3 | 67.8 | 90.2 | 61.9 | 21.9 | **96.4** |
| | XSum | 95.8 | 74.9 | 89.2 | 91.6 | 86.6 | 99.1 | 91.5 | 9.7 | **99.9** |
| | WP-s | 97.4 | 80.7 | 94.7 | 89.6 | 94.2 | 99.2 | 70.9 | 27.5 | **99.8** |
| GPT4 | PubMed | 81.0 | 59.7 | 68.1 | 68.1 | 63.3 | 85.0 | 53.1 | 28.1 | **96.1** |
| | XSum | 79.8 | 66.4 | 67.1 | 74.5 | 64.8 | 90.7 | 67.8 | 50.3 | **98.7** |
| | WP-s | 85.5 | 71.5 | 80.9 | 70.3 | 78.0 | 96.1 | 50.7 | 45.3 | **98.8** |
| GPT4 Turbo | PubMed | 86.5 | 60.8 | 63.6 | 73.5 | 63.7 | 88.8 | 55.8 | 31.0 | **97.0** |
| | XSum | 90.9 | 73.4 | 83.2 | 87.9 | 81.8 | 97.4 | 88.2 | 4.4 | **100.0** |
| | WP-s | 97.6 | 80.8 | 92.8 | 92.9 | 92.5 | 99.4 | 72.3 | 22.5 | **99.8** |
| Llama3 70B | PubMed | 85.4 | 60.9 | 66.0 | 71.3 | 65.0 | 90.8 | 52.9 | 35.1 | **96.3** |
| | XSum | 97.9 | 74.9 | 93.2 | 95.5 | 93.8 | 99.7 | 96.2 | 7.1 | **99.8** |
| | WP-s | 97.1 | 77.9 | 95.5 | 90.1 | 95.8 | **99.9** | 77.5 | 28.1 | 99.5 |
| Gemini pro | PubMed | 83.0 | 58.3 | 63.2 | 75.0 | 66.8 | 82.1 | 57.3 | 39.3 | **86.4** |
| | XSum | 78.6 | 44.5 | 72.8 | 73.0 | **79.6** | 79.5 | 72.2 | 54.7 | 74.5 |
| | WP-s | 75.8 | 63.0 | 77.8 | 72.7 | 81.1 | 78.0 | 70.2 | 48.0 | **86.4** |
| Calude3 Opus | PubMed | 85.5 | 60.3 | 66.3 | 74.3 | 64.4 | 88.2 | 48.9 | 33.1 | **96.4** |
| | XSum | 95.9 | 71.1 | 85.3 | 89.7 | 84.7 | 96.2 | 86.2 | 5.3 | **99.9** |
| | WP-s | 93.8 | 75.0 | 91.9 | 86.5 | 91.8 | 93.5 | 65.7 | 24.1 | **99.5** |
| Average | - | 88.6 | 67.4 | 79.2 | 80.6 | 78.7 | 91.9 | 68.9 | 28.6 | **95.8** |

(b) MGTBench [15]: Essay, Reuters, and WritingPrompts (WP)

| Model | Domain | Loglik. | Rank | D-GPT | LRR | NPR | FD-GPT | Open-D | Chat-D | Ours |
|---|---|---|---|---|---|---|---|---|---|---|
| GPT3.5 Turbo | Essay | 97.3 | 95.7 | 57.8 | 97.8 | 48.1 | 99.6 | 57.5 | 81.5 | **100.0** |
| | Reuters | 98.2 | 94.8 | 50.5 | 98.7 | 51.1 | **99.9** | 98.5 | 97.2 | 99.9 |
| | WP | 89.8 | 90.2 | 52.9 | 77.2 | 48.3 | 91.7 | 50.8 | 66.3 | **100.0** |
| GPT4 Turbo | Essay | 96.5 | 93.9 | 58.9 | 93.9 | 62.4 | 98.9 | 55.8 | 77.1 | **99.9** |
| | Reuters | 95.8 | 93.1 | 52.6 | 94.9 | 53.3 | 99.4 | 87.5 | 92.4 | **99.9** |
| | WP | 94.2 | 91.0 | 53.5 | 85.2 | 55.3 | 93.0 | 68.2 | 67.9 | **99.9** |
| Llama3 70B | Essay | 98.3 | 95.3 | 56.2 | 98.9 | 57.8 | 99.5 | 83.9 | 91.7 | **100.0** |
| | Reuters | 99.9 | 89.7 | 58.9 | 98.7 | 59.2 | **100.0** | 96.7 | 90.8 | **100.0** |
| | WP | 97.3 | 90.8 | 57.2 | 91.1 | 60.4 | 99.1 | 86.6 | 77.3 | **99.8** |
| Gemini pro | Essay | 98.3 | 93.6 | 64.4 | 97.7 | 65.5 | 98.3 | 48.9 | 65.9 | **100.0** |
| | Reuters | 99.9 | 83.1 | 73.0 | 99.3 | 74.9 | **100.0** | 95.3 | 91.5 | **100.0** |
| | WP | 91.7 | 82.0 | 63.9 | 76.7 | 67.3 | 99.2 | 68.8 | 73.4 | **99.8** |
| Claude | Essay | 91.6 | 85.9 | 44.2 | 82.7 | 48.7 | 83.6 | 32.4 | 19.6 | **99.7** |
| | Reuters | 91.3 | 79.5 | 68.1 | 79.2 | 68.7 | 87.8 | 65.5 | 25.6 | **99.8** |
| | WP | 88.4 | 80.0 | 60.0 | 71.2 | 60.7 | 74.1 | 46.2 | 26.7 | **99.1** |
| Average | - | 95.2 | 89.2 | 58.1 | 89.5 | 58.8 | 94.9 | 69.5 | 69.7 | **99.9** |

## 4.1 Main Results

In Table 2, we show the LGT detection performance of ReMoDetect and other detection baselines. Overall, ReMoDetect significantly outperforms prior detection methods by a large margin, achieving state-of-the-art performance in average AUROC. For instance, on the Fast-DetectGPT benchmark, ReMoDetect improves the prior best average AUROC from 91.9%→95.8%. Moreover, it is worth noting that the improvement is consistent in MGTbench, indicating the generalization ability of ReMoDetect, despite the fact that it's trained on specific LGTs (i.e., ChatGPT texts from HC3). Thus, we believe the continual preference tuning with replay indeed helped prevent the overfitting.

Table 3: Comparison with ReMoDetect (Ours) and GPTZero [39], a commercial black-box LGT detection API. We report the average AUROC (%) on the Fast-DetectGPT benchmark, including PubMed, XSum, and WritingPrompts. The bold indicates the best results.

| Model | GPT 3.5 Turbo | GPT4 | GPT4 Turbo | Llama3 70B | Gemini pro | Claude3-Opus |
|---|---|---|---|---|---|---|
| GPTZero | 93.5 | 88.5 | 95.7 | 96.6 | **82.9** | 95.7 |
| **Ours** | **98.7** | **97.9** | **98.9** | **98.5** | 82.4 | **98.6** |

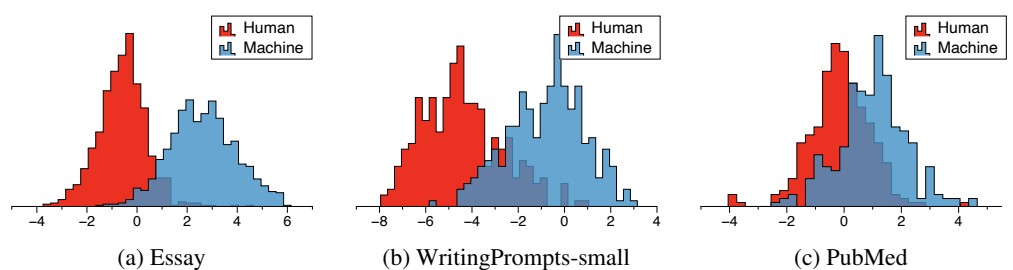

(a) Essay       (b) WritingPrompts-small       (c) PubMed

Figure 3: Predicted reward distribution of human written texts and LGTs on three different domains, including (a) Essay, (b) WritingPrompts-small, and (c) PubMed. We use the reward model from OpenAssistant [3]. 'Machine' denotes GPT4 Turbo and Claude3 Opus generated texts.

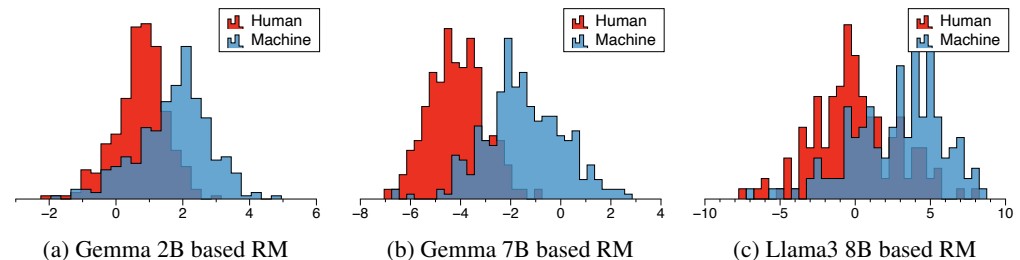

(a) Gemma 2B based RM       (b) Gemma 7B based RM       (c) Llama3 8B based RM

Figure 4: Predicted reward distribution of human-written texts and LGTs on three different reward models (RMs), including (a) Gemma 2B (b) Gemma 7B, and (c) Llama3 8B. 'Machine' denotes GPT4 Turbo and Claude3 Opus generated texts. We use WritingPrompts-small as the text domain.

**Comparison with a commercial detection method.** We also compare ReMoDetect with a commercial LGT detection method, GPTZero, under the Fast-DetectGPT benchmark. Somewhat interestingly, as shown in Table 3, ReMoDetect significantly outperforms GPTZero in all considered aligned LLMs except for one in terms of the average AUROC. It is worth noting that ReMoDetect only has seen ChatGPT datasets and partially rephrased texts by Llama3 70B, indicating the rest of the aligned LLMs are unseen distribution to ReMoDetect. We believe further improving the performance of ReMoDetect by enlarging the training corpus using more aligned LLM will be an interesting future direction to explore, showing an impact on the open-source community.

## 4.2 Reward Model Analysis

**More observation studies.** In addition to our observation study presented in Table 1 and Figure 1, we considered (i) more text domains and (ii) different types of reward models to rigorously verify our observation (i.e., aligned LLMs generate texts with higher predicted preference compared to human-written texts). To this end, we use a pre-trained reward model without further fine-tuning. First, we show that our observation is consistent across multiple text domains (in Figure 3). Interestingly, the predicted reward separation between LGTs and human-written texts is more significant in Essay and WritingPrompts-small compared to PubMed (i.e., a biology expert written data), possibly implying that alignment training is done more on relatively common texts compared to expert datasets. Second, we also observed that LGTs have higher preference compared to human-written texts on other reward models as well (in Figure 4). Intriguingly, a larger reward model within the same model family (i.e., Gemma 7B compared to 2B) shows better separation of the predicted score, showing the possibility of ReMoDetect's scaling law, i.e., using a large reward model will improve the detection performance. We also provide more results of our observation studies in Appendix B.5.

Table 4: Robustness against rephrasing attacks. We report the average AUROC (%) before ('Original') and after ('Attacked') the rephrasing attack with T5-3B on the Fast-DetectGPT benchmark, including XSum, PubMed, and small-sized WritingPrompts. Values in the parenthesis indicate the relative performance drop after the rephrasing attack. The bold indicates the best result.

| Model | Accuracy | Loglik. | D-GPT | NPR | FD-GPT | **Ours** |
|-------|----------|---------|-------|-----|--------|----------|
| GPT3.5 | Original | 93.6 | 86.1 | 82.9 | 96.1 | **98.7** |
| Turbo | Attacked | 80.5 (-14.0%) | 60.3 (-30.0%) | 73.5 (-11.3%) | 87.2 (-9.3%) | **91.4 (-7.4%)** |
| GPT4 | Original | 91.7 | 79.9 | 79.4 | 95.2 | **98.9** |
| Turbo | Attacked | 80.0 (-12.7%) | 50.3 (-37.0%) | 61.3 (-22.8%) | 87.3 (-8.3%) | **94.6 (-4.4%)** |
| Claude3 | Original | 91.7 | 81.1 | 80.3 | 92.6 | **98.6** |
| Opus | Attacked | 80.5 (-15.8%) | 55.2 (-32.0%) | 60.1 (-25.2%) | 81.6 (-11.9%) | **91.1 (-7.1%)** |

**Reward distribution change after training.**
We additionally analyze the predicted reward distribution change made by our training objective Eq (2). To this end, we visualize the reward distribution before and after the training the reward model by using GPT4-Turbo generated texts on Eassy domain. As shown in Figure 5, our training objective indeed increases the separation of the predicted reward distribution between human-written texts and LGTs. Interestingly, the LGT's reward distribution becomes more compact and equally higher, whereas the reward distribution of human-written texts becomes more dispersed.

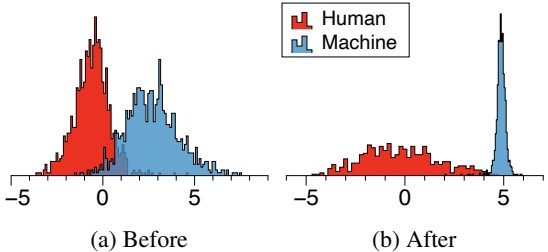

(a) Before          (b) After

Figure 5: Predicted reward distribution of human written texts and LGTs (a) 'Before' and (b) 'After' training the reward model with Eq (2). 'Machine' denotes GPT4-Turbo generated texts on Eassy domain.

We conjecture that this difference arises because human-written texts are produced by diverse individuals with varying backgrounds and experiences, while aligned LLMs share somewhat similar training receipts across models.

### 4.3 Additional Analysis

In this section, we provide more analysis of ReMoDetect. Here, we mainly consider baselines that show effectiveness in the main experiment (e.g., Fast-DetectGPT in Table 2) and consider the GPT4 family and Claude3 as aligned LLMs.

**Robustness to unseen distributions.**
We verify the claim that training detectors on specific LGTs may introduce bias and require careful training by showing the failure cases of the prior work and the robustness of ReMoDetect to unseen distributions. To this end, we compare ReMoDetect with ChatGPT-Detector, which is trained on the same

Table 5: AUROC (%) of ChatGPT-D and ReMoDetect (ours), on datasets and models that are seen (**S**) or unseen (**U**) during training time. The bold denotes the best results.

| Domain Model | HC3 (**S**) GPT3.5 (**S**) | HC3 (**S**) Claude3 (**U**) | WP-s (**U**) Claude3 (**U**) |
|--------------|------|------|------|
| ChatGPT-D | 99.8 | 96.7 | 24.1 |
| **Ours** | **99.9** | **99.9** | **99.5** |

dataset (i.e., GPT3.5 Turbo generated texts on the HC3 domain) and evaluate on the unseen domain (i.e., WritingPrompts-small) and machine (i.e., Claude3 Opus). As shown in Table 5, both ReMoDetect and ChatGPT-Detector work well on the seen domain and LLM, while ReMoDetect shows significant robustness to unseen distributions compared to ChatGPT-Detector. For instance, the AUROC of ChatGPT-Detector in the seen domain dropped from 99.8%→24.1% when tested on the unseen domain while ReMoDetect retains the original accuracy, i.e., 99.9%→99.5%.

**Robustness against rephrasing attacks.** One possible challenging scenario is detecting the rephrased texts by another LM (known as rephrasing attacks) [42], i.e., first generate texts with powerful LLMs and later modify them with another LLM. To this end, we follow the prior work by using a T5-3B specifically trained for rephrasing attack [42]. As shown in Table 4, ReMoDetect significantly and consistently outperforms all baselines. It is worth noting that our relative drop in performance is also significantly lower than other baselines, indicating strong robustness of ReMoDetect.

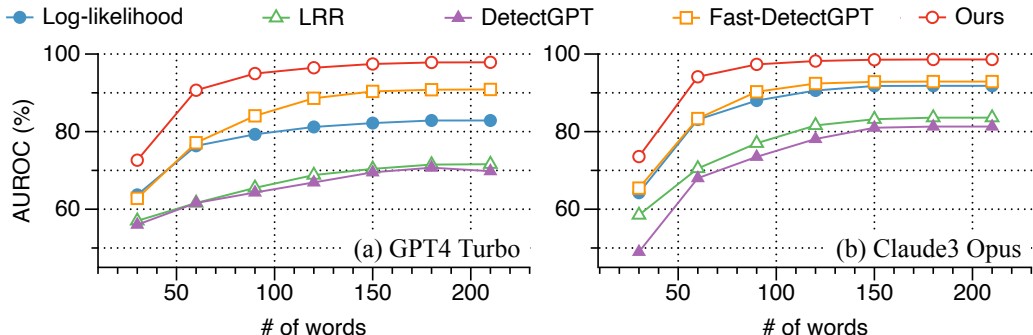

Figure 6: Average AUROC (%) of various LGT detection methods on various input response lengths by monotonically increasing 30 words each. We consider three text domains from the Fast-DetectGPT benchmark and two aligned LLM, including (a) GPT4 Turbo and (b) Claude3 Opus.

Table 6: LGT Detection results on non-RLHF trained LLMs. We report AUROC (%) of multiple LGT detection methods, including log-likelihood (Loglik.), Rank, Fast-DetectGPT (FD-GPT), OpenAI-Detector (Open-D), ChatGPT-Detector (Chat-D), and ReMoDetect (Ours). We consider LGT detection benchmarks from Fast-DetectGPT: PubMed, XSum, and WritingPrompts-small (WP-s). Here, Phi-3 medium is DPO trained and OLMo-7B-SFT is SFT-only trained. The bold indicates the best result within the group.

| Model | Domain | Loglik. | Rank | FD-GPT | Open-D | Chat-D | **Ours** |
|-------|--------|---------|------|--------|--------|--------|----------|
| Phi-3 mini | PubMed | 65.0 | 56.2 | 63.7 | 37.7 | 80.7 | **94.5** |
| | XSum | 70.3 | 64.1 | 91.0 | 82.7 | 23.4 | **97.6** |
| | WP-s | 82.4 | 73. | 96.7 | 60.0 | 31.1 | **99.3** |
| Phi-3 small | PubMed | 57.2 | 50.4 | 59.9 | 31.9 | 82.7 | **91.7** |
| | XSum | 81.1 | 69.7 | 95.6 | 79.3 | 19.5 | **98.7** |
| | WP-s | 84.0 | 72.3 | 97.2 | 58.6 | 32.2 | **97.4** |
| Phi-3 medium | PubMed | 65.4 | 55.4 | 61.7 | 34.2 | 15.8 | **95.2** |
| | XSum | 64.5 | 61.2 | 85.4 | 75.0 | 18.1 | **98.0** |
| | WP-s | 83.1 | 73.6 | 95.7 | 53.9 | 38.5 | **98.8** |
| OLMo 7B-SFT | PubMed | 88.4 | 60.5 | 92.8 | 62.0 | 23.6 | **94.1** |
| | XSum | 96.6 | 66.0 | **99.1** | 97.3 | 5.9 | 98.1 |
| | WP-s | 98.1 | 78.5 | 98.8 | 95.2 | 19.5 | **99.2** |
| Average | - | 86.0 | 63.8 | 91.2 | 72.2 | 43.8 | **95.3** |

**Robustness on input response length.** By following the prior work [13], we also measure the robustness of ReMoDetect on the input response length (i.e., # of words in $y$). Note that shorter responses are hard to detect as there is less evidence to identify the characteristics of humans and LLMs. As shown in Figure 6, ReMoDetect significantly outperforms the major baselines. Interestingly, our method can even outperform the best baseline with 71.4% fewer words, showing significant robustness on short input responses. For instance, Fast-DetectGPT reaches AUROC of 91.8% with 210 words, while ReMoDetect reaches 94.1% with 60 words under Claude3 Opus.

**ReMoDetect for non-RLHF aligned LLMs.** We additionally consider aligned LLMs that do not use reward models for alignment training, i.e., non-RLHF trained LLMs. To this end, we consider aligned LLMs that use Direct Preference Optimization (DPO) [30], an alternative alignment training to RLHF. Note that a recently released Phi-3 [25] only uses DPO (followed by supervised fine-tuning; SFT) for alignment training and shows remarkable performance in various domains, thus being considered an aligned LLM in our experiment. As shown in Table 6, ReMoDetect also outperforms baselines in all cases, showing that our method can be applicable even if aligned LLMs are not trained with reward models. Furthermore, we also considered the detection scenario for the SFT-only model that does not use the alignment training. Here, we observe that ReMoDetect effectively detects the LGTs from the SFT-only model as well as outperforming other baselines. We believe this is because the SFT implicitly trains the model to reflect the human preference from the instruction tuning dataset [10], thus making the ReMoDetect well-detect the texts from SFT models.

Table 7: Contribution of each proposed component of ReMoDetect on detecting aligned LGTs from human-written texts. We report the average detection performance of GPT4 under text domains in the Fast-DetectGPT benchmark. All values are percentages, and the best results are indicated in bold.

| Continual Fine-tuning (No Replay) | with Replay Buffers | Mixed Text Reward Modeling | AUROC | AUPR | TPR at FPR 1% |
|---|---|---|---|---|---|
| - | - | - | 79.0 | 79.2 | 16.7 |
| ✓ | - | - | 90.5 | 91.0 | 38.9 |
| ✓ | ✓ | - | 95.5 | 95.8 | 59.3 |
| ✓ | ✓ | ✓ | **97.9** | **98.0** | **77.0** |

Table 8: Comparison of detection time, model parameters, and average AUROC (%) of Fast-DetectGPT benchmark for various LGT detection methods. Detection time was measured in an A6000 GPU, and the overall detection time was measured for 300 XSum dataset samples.

| Method | Detection Time (secs) | Model Parameters | AUROC |
|---|---|---|---|
| Log-likelihood | 11.7 | 2.7B | 88.6 |
| DetectGPT | 7738.8 | 3B & 2.7B | 79.2 |
| NPR | 7837.3 | 3B & 2.7B | 78.7 |
| Fast-DetectGPT | 62.7 | 6B & 2.7B | 91.9 |
| Ours | **8.7** | **0.5B** | **95.8** |

**Component analysis.** We perform an analysis on each component of our method in detecting GPT4 generated texts: namely, the use of (i) continual fine-tuning with no replay $\lambda = 0$, (ii) the replay buffers, and (iii) the reward modeling with Human/LLM mixed texts, by comparing multiple detection performance metrics. Results in Table 7 show each component is indeed important, where gradually applying our techniques shows a stepwise significant improvement.

**Inference time efficiency.** In Table 8, we compared detection time, model parameter size, and average AUROC on the Fast-DetectGPT benchmark. The detection time was measured in an A6000 GPU, and the overall detection time was measured with 300 samples of the human/GPT3.5 Turbo XSum dataset. ReMoDetect shows the best average AUROC performance among the methods, but 7.2 times faster, and uses a 17.4 times smaller model than the second best model, Fast-DetectGPT.

## 5 Discussion and Conclusion

We propose ReMoDetect, a novel and effective LLM-generated text (LGT) detection framework. Based on the novel observation that the reward model well recognizes LGTs from human-written texts, we continually fine-tune the reward model to further separate reward scores of two distributions while preventing the overfitting bias using the replay technique. Furthermore, we suggest a Human/LLM mixed text dataset for reward modeling, learning a better decision boundary of the reward model detector. Experimental results further demonstrate that ReMoDetect significantly improves the prior state-of-the-art results in detecting aligned LGTs.

**Future works and limitations.** We believe it will be an interesting future direction to train LLMs using the reward model of ReMoDetect. Making the predictive reward distribution of LGTs more well-spread (like the human-written texts in Figure 5), can be a step toward making LLMs more human-like. Additionally, a potential limitation of ReMoDetect is the somewhat lack of accessibility of reward models. While there are some open-source reward models available (that we have used throughout the paper), their number is still limited compared to open-source LLMs. We believe that as the open-source community grows and more pre-trained reward models (or human preference datasets) become available, ReMoDetect will be improved further.

**Societal impact.** This paper presents ReMoDetect that improves the performance of detecting aligned LGTs. We expect that our approach will show numerous positive impacts by detecting LGTs, such as in fake news and academic plagiarism. One possible negative impact can be the improved adversarial mechanism followed by the improved detection method (i.e., ReMoDetect); thus, incorporating such a scenario will be an interesting future direction to explore, where we believe using ReMoDetect to such a scenario can be promising (as it shows robustness in multiple cases in Section 4.3).

## Acknowledgements

We thank Jongheon Jeong and Myungkyu Koo for providing helpful feedback and suggestions in preparing an earlier version of the manuscript. This work was supported by Institute for Information & communications Technology Promotion(IITP) grant funded by the Korea government(MSIT) (No.RS-2019-II190075 Artificial Intelligence Graduate School Program(KAIST)) and NIPA(National IT Industry Promotion Agency), through the Ministry of Science and ICT (Hyperscale AI flagship project).

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

# Appendix

## A    Experimental Details

In this section, we describe the experimental details of Section 4, including ReMoDetect and baselines.

### A.1    Dataset Details

In this section, we describe the dataset we used in training and evaluation. Also, explain how we generated the additional datasets.

- **HC3.** HC3 is a question-and-answering dataset that consists of answers written by humans and generated by ChatGPT corresponding to the same questions. The dataset is a collection of several domains: reddit_eli5, open_qa, wiki_csai, medicine, and finance. We used training samples of 2,200 and validation samples of 1,000, which is the same subset of HC3 as the prior work [6, 40]. We used the filtered version of the HC3 dataset.

- **Reuters.** Reuters is a news dataset that consists of news articles written by humans and generated by LLM corresponding to the same subjects. We brought the dataset from MGTBench [15] and followed the construction recipe to generate more evaluation datasets for recent LLMs. The dataset comprises 1,000 news articles written by humans and generated by LLM, GPT3.5 Turbo, GPT4 Turbo, Claude, Claude Opus, Llama3 70B instruct. GPT3.5 Turbo and Claude dataset is from MGTBench [15]. We made the same evaluation set for Essay and WritingPrompts.

- **Essay.** Essay consists of essays extracted from IvtPandas. We brought the dataset from MGT-Bench [15] and followed the construction recipe to generate more evaluation datasets for recent LLMs. The dataset consists of diverse essay subjects across various academic disciplines. The dataset comprises 1,000 samples of Essays written by humans and generated by aligned LLMs.

- **WritingPrompts.** WritingPrompts is the creative writing prompt shared on r/WritingPrompts of Reddit. We brought the dataset from MGTBench [15] and followed the construction recipe to generate more evaluation datasets for recent LLMs. The dataset comprises 1,000 samples of WritingPrompts written by humans and generated by LLMs.

- **WritingPrompts-small.** WritingPrompts-small is the creative writing prompt shared on Reddit r/WritingPrompts. We brought the dataset from FastDetectGPT [13] and followed the construction recipe to generate more evaluation datasets for recent LLMs. The dataset comprises 150 samples of WritingPrompts written by humans and generated by LLM.

- **XSum.** Xsum is a news dataset comprising news articles written by humans and generated by LLM corresponding to the same subjects. We brought the dataset from FastDetectGPT [13] and followed the construction recipe to generate more evaluation datasets for recent LLMs. The dataset comprises 150 news articles written by humans and generated by LLMs.

- **PubMeds.** PubMed is a question-and-answering dataset of biomedical research domains written by humans and generated by LLMs corresponding to the questions. We brought the dataset from FastDetectGPT [13] and followed the construction recipe to generate more evaluation datasets for recent LLMs. The dataset comprises 150 QA pairs written by humans and generated by LLMs.

- **Human/LLM mixed datasets.** We rephrase the human-written text from the HC3 dataset using Llama3 70B instruct [41]: We first select 50% of the indices in the paragraph, then rephrase selected sentences using the following prompt to the rephrasing LLM:

> Please paraphrase sentence numbers <idxlist> in given written texts.
> ...
> <ith> sentence: <xxx>
> <i+1th> sentence: <xxx>
> ...

The <idxlist> is a 50% randomly selected index list of sentences like "[0,2,5,7]", Then list all the sentences of the passages like "<5th> sentence: A fellow high school student, typically a 3 or 4 - there's a lot of stress involved."

## A.2 Aligned LLM Spec Details

The API version of our dataset is as follows:

- OpenAI / GPT3.5 Turbo : `gpt-3.5-turbo-0301`

- OpenAI / GPT4 : `gpt-4`

- OpenAI / GPT4 Turbo : `gpt-4-turbo-2024-04-09`

- Anthropic / Claude3 Opus : `claude-3-opus-20240229`

- Anthropic / Claude3 Sonnet : `claude-3-sonnet-20240229`

- Anthropic / Claude3 Haiku : `claude-3-haiku-20240307`

- Google / Gemini pro : `gemini-pro 2024-02-01`

We use the open-source model for Llama3 70B instruct[3] and Phi-3 [25]. Here, we use Phi-3 with a 4K context length for mini[4] and medium[5], whereas we use an 8K context length for Phi-3 small[6] (Phi-3 small only has 8K model). We spent $56.0 for OpenAI API and $156.6 for Anthropic API.

## A.3 Training and Evaluation Details

**Training details of ReMoDetect.** We use AdamW optimizer with a learning rate of $2.0 \times 10^{-5}$ with 10% warm up and cosine decay and train it for one epoch. For the $\lambda$ constant for regularization using replay buffer, we used $\lambda = 0.01$. For the $\beta_1, \beta_2$ parameters that choose the contribution of the mixed data, we used 0.3 and 0.3. As for the replay buffer datasets, we use 'Anthropic/hh-rlhf'[7] and 'Dahoas/synthetic-instruct-gptj-pairwise'[8] from the huggingface datasets library as our base reward model [3] used these datasets for training. We use the same batch size for the training sample and replay buffer sample, which ends up with a total batch size of four.

**Reward model details.** We mainly used the open-source reward model from OpenAssistant [9], which is based on DeBERTa-v3-Large [14]; the model parameter size is 435M and trained with a human preference dataset. Additionally, in Figure 4, we used other reward models, weqweasdas/RM-Gemma-2B[10], weqweasdas/RM-Gemma-7B[11], and sfairXC/FsfairX-LLaMA3-RM-v0.1[12] from the huggingface library in order to verify our observations in other reward models.

**Detection metrics.** For the evaluation, we measure the following metrics to verify the effectiveness of the detection methods in distinguishing human-written texts and LGTs.

- **True positive rate (TPR) at 1% false positive rate (FPR)**. Let TP, TN, FP, and FN denote true positive, true negative, false positive, and false negative, respectively. We measure TPR = TP / (TP+FN) when FPR = FP / (FP+TN) is 1%.

- **Area under the receiver operating characteristic curve (AUROC).** The ROC curve is a graph plotting TPR against the false positive rate = FP / (FP+TN) by varying a threshold.

- **Area under the precision-recall curve (AUPR).** The PR curve is a graph plotting the precision = TP / (TP+FP) against recall = TP / (TP+FN) by varying a threshold.

**Resource Details.** For the main development, we mainly use Intel(R) Xeon(R) Gold 6426Y CPU @ 2.50GHz and a single A6000 48GB GPU.

---

[3]`https://huggingface.co/meta-llama/Meta-Llama-3-70B-Instruct`
[4]`https://huggingface.co/microsoft/Phi-3-mini-4k-instruct`
[5]`https://huggingface.co/microsoft/Phi-3-medium-4k-instruct`
[6]`https://huggingface.co/microsoft/Phi-3-small-8k-instruct`
[7]`https://huggingface.co/Anthropic/hh-rlhf`
[8]`https://huggingface.co/Dahoas/synthetic-instruct-gptj-pairwise`
[9]`https://huggingface.co/OpenAssistant/reward-model-deberta-v3-large-v2`
[10]`https://huggingface.co/weqweasdas/RM-Gemma-2B`
[11]`https://huggingface.co/weqweasdas/RM-Gemma-7B`
[12]`https://huggingface.co/sfairXC/FsfairX-LLaMA3-RM-v0.1`

### A.4 Robustness Evaluation Details

**Rephrasing attack.** To check the robustness of our method against rephrasing attacks, we utilized T5-3B-based paraphraser [42] to paraphrase the sentences in the passage. We conducted experiments with hyper-parameters to max_length = 256, top_k = 200, top_p = 0.95. The result is in Table 4.

**Input response length.** To check the robustness of our method against input response length, we truncated the given test dataset to various word lengths. First, we tokenized the given paragraph into words using the nltk framework. Then, we truncate each passage into target word lengths. We tested for word length $\in [30, 60, 90, 120, 150, 180, 210]$. The result is in Figure 6.

### A.5 Baseline Details

We describe baselines that we compared with ReMoDetect in Fast-DetectGPT benchmark [13] and MGTBench [15]. We use implementations and backbone models introduced in Fast-DetectGPT [13].

- **Log-likelihood**, **Rank** [20]. These methods use LLM to measure the token-wise log probability and rank of the words, then average the metric of each token to generate a score for the text. For the baseline experiments, we utilized GPT-neo-2.7B as their base model.

- **DetectGPT** [13], **NPR** [21]. DetectGPT, NPR is designed to measure changes in a model's log probability and log-rank function when slight perturbations are introduced to the original text. For the baseline experiments, we utilized GPT-neo-2.7B as their base model and T5-3B for paraphrasing, and we perturbed 100 for each paragraph.

- **LRR** [21]. LRR used the Log-likelihood log-rank Ratio, which merges the benefits of log-likelihood and log-rank. We utilized GPT-neo-2.7B as their base model.

- **Fast-DetectGPT** [13]. Fast-DetectGPT shares the same spirt as DetectGPT, where it uses the conditional probability function by sampling the text using the base model instead of perturbation using T5 models, thus showing efficiency. Following the original paper setting, we used GPT-J as a base model and GPT-neo-2.7B as a scoring model.

- **OpenAI-Detector** [20]. OpenAI-Detector is a RoBERTa-based supervised finetuned model trained with pairs of human-written and GPT2-generated texts.

- **ChatGPT-Detector** [6]. ChatGPT-Detector is a RoBERTa-based supervised finetuned model trained with the HC3 dataset, which consists of human-written and ChatGPT generated texts.

## B  Additional Experimental Results

### B.1 Comparison with Additional Baselines

Table 9: AUROC(%) on MGT benchmark[15] for different baselines: Log Rank [13], Entropy [34], and GLTR [34]. The bold indicated the best result.

| Model | Domain | GPT 3.5 Turbo | GPT4 Turbo | Llama3 70B | Gemini pro | Claude |
|---|---|---|---|---|---|---|
| Log Rank [13] | Essay | 98.1 | 96.7 | 98.7 | 97.9 | 89.1 |
| | Reuters | 98.6 | 95.8 | 99.7 | 99.7 | 85.5 |
| | WP | 86.5 | 90.5 | 95.3 | 87.6 | 79.9 |
| Entropy [34] | Essay | 94.1 | 90.2 | 91.9 | 89.0 | 84.1 |
| | Reuters | 77.8 | 75.5 | 78.6 | 78.3 | 77.9 |
| | WP | 84.0 | 85.4 | 82.0 | 64.1 | 80.9 |
| GLTR [34] | Essay | 97.8 | 95.9 | 98.7 | 97.8 | 87.1 |
| | Reuters | 98.4 | 94.8 | 99.5 | 99.6 | 84.7 |
| | WP | 85.9 | 88.4 | 95.2 | 85.9 | 79.1 |
| **ReMoDetect** | Essay | **100.0** | **99.9** | **100.0** | **100.0** | **99.7** |
| | Reuters | **99.9** | **99.9** | **100.0** | **100.0** | **99.8** |
| | WP | **100.0** | **99.9** | **99.8** | **99.8** | **99.1** |

In Table 9, we compare other baselines Log Rank [13], Entropy [34], GLTR [34], and ReMoDetect on MGT benchmark. ReMoDetect consistently outperforms other baselines in MGT benchmark.

## B.2 Comparison on Additional Aligned LLMs

Table 10: AUROC(%) on Fast-DetectGPT benchmark [13] for different models: Claude3 Haiku [5] and Sonnet [5]. The bold indicates the best result.

| Model | Domain | Loglik. | Rank | D-GPT | LRR | NPR | FD-GPT | Open-D | Chat-D | **Ours** |
|---|---|---|---|---|---|---|---|---|---|---|
| Claude3 Haiku | PubMed | 87.0 | 60.9 | 67.5 | 75.5 | 66.9 | 90.9 | 56.2 | 28.3 | **96.3** |
| | XSum | 96.2 | 73.8 | 91.9 | 93.0 | 90.6 | **99.8** | 93.9 | 6.8 | **99.8** |
| | WP-s | 98.2 | 78.8 | 94.1 | 93.1 | 94.8 | 99.7 | 82.4 | 27.9 | **99.8** |
| Claude3 Sonnet | PubMed | 84.4 | 60.6 | 64.9 | 71.8 | 64.5 | 86.5 | 52.4 | 31.0 | **96.4** |
| | XSum | 90.1 | 70.9 | 84.4 | 86.2 | 84.1 | 94.7 | 76.0 | 13.7 | **98.7** |
| | WP-s | 94.9 | 77.7 | 93.5 | 87.5 | 93.2 | 98.0 | 57.1 | 35.6 | **99.7** |

In Table 10, we evaluate Claude3 Haiku and Claude3 Sonnet, which are serviced by Anthropic and are smaller versions of Claude3 Opus. ReMoDetect consistently outperforms other baselines in the evaluation, demonstrating that our detector can detect these smaller models effectively.

## B.3 Additional Performance Metric

Table 11: TPR(%) at FPR 1% and AUPR (%) of multiple LLM-generated text detection methods, including log-likelihood (Loglik.) [20], Rank [20], DetectGPT (D-GPT) [11], LRR [21], NPR [21], Fast-DetectGPT (FD-GPT) [13], OpenAI-Detector (Open-D) [20], ChatGPT-Detector (Chat-D) [6], and ReMoDetect (Ours). We consider LLM-generated text detection benchmarks from Fast-DetectGPT [13]. The bold indicates the best result within the group.

(a) TPR at FPR 1%

| Model | Domain | Loglik. | Rank | D-GPT | LRR | NPR | FD-GPT | Open-D | Chat-D | **Ours** |
|---|---|---|---|---|---|---|---|---|---|---|
| GPT3.5 Turbo | PubMed | 10.7 | 4.0 | 0.0 | 8.0 | 5.3 | 44.0 | 2.0 | 1.3 | **63.3** |
| | XSum | 68.7 | 12.7 | 25.3 | 47.3 | 15.3 | 82.0 | 46.0 | 0.0 | **96.7** |
| | WP-s | 64.7 | 13.3 | 28.0 | 28.7 | 37.3 | 87.3 | 9.3 | 0.0 | **97.3** |
| GPT4 | PubMed | 8.7 | 3.3 | 0.0 | 6.0 | 5.3 | 18.0 | 2.7 | 1.3 | **70.0** |
| | XSum | 24.0 | 1.3 | 1.3 | 11.3 | 6.7 | 32.7 | 13.3 | 0.0 | **79.3** |
| | WP-s | 9.3 | 2.7 | 10.7 | 2.7 | 2.0 | 44.0 | 1.3 | 0.0 | **82.0** |
| GPT4 Turbo | PubMed | 12.7 | 4.7 | 0.7 | 13.3 | 4.7 | 27.3 | 0.7 | 0.0 | **67.3** |
| | XSum | 46.3 | 8.8 | 9.5 | 46.3 | 10.9 | 68.0 | 42.9 | 0.0 | **99.3** |
| | WP-s | 60.4 | 18.8 | 15.4 | 41.6 | 34.2 | 80.5 | 11.4 | 0.0 | **98.7** |
| Calude3 Opus | PubMed | 14.0 | 5.3 | 0.7 | 12.0 | 4.0 | 26.0 | 1.3 | 0.7 | **62.7** |
| | XSum | 42.7 | 11.3 | 26.7 | 44.7 | 24.0 | 75.3 | 43.3 | 0.0 | **97.3** |
| | WP-s | 54.7 | 16.7 | 37.3 | 24.0 | 55.3 | 76.7 | 8.0 | 0.7 | **96.0** |

(b) AUPR

| Model | Domain | Loglik. | Rank | D-GPT | LRR | NPR | FD-GPT | Open-D | Chat-D | **Ours** |
|---|---|---|---|---|---|---|---|---|---|---|
| GPT3.5 Turbo | PubMed | 86.5 | 62.8 | 55.1 | 73.7 | 62.4 | 90.8 | 61.5 | 36.5 | **96.9** |
| | XSum | 95.3 | 77.1 | 88.2 | 91.6 | 85.5 | 99.2 | 93.4 | 32.0 | **99.9** |
| | WP-s | 97.7 | 81.6 | 94.0 | 89.3 | 94.1 | 99.3 | 71.0 | 37.8 | **99.8** |
| GPT4 | PubMed | 79.9 | 60.5 | 54.7 | 67.0 | 59.7 | 84.4 | 55.5 | 38.8 | **96.7** |
| | XSum | 80.1 | 65.4 | 63.2 | 75.6 | 62.5 | 91.1 | 73.8 | 58.6 | **98.7** |
| | WP-s | 81.6 | 68.1 | 79.4 | 66.1 | 74.1 | 96.0 | 50.2 | 46.5 | **98.7** |
| GPT4 Turbo | PubMed | 85.0 | 62.8 | 59.1 | 74.5 | 61.6 | 89.4 | 56.1 | 38.6 | **97.4** |
| | XSum | 91.5 | 75.7 | 81.3 | 89.7 | 81.4 | 97.6 | 90.8 | 31.0 | **100.0** |
| | WP-s | 97.6 | 82.9 | 91.5 | 93.1 | 92.5 | 99.4 | 74.0 | 36.6 | **99.8** |
| Calude3 Opus | PubMed | 84.6 | 62.7 | 60.1 | 74.4 | 60.0 | 88.4 | 52.1 | 40.4 | **96.7** |
| | XSum | 93.8 | 74.2 | 85.5 | 90.7 | 85.0 | 96.9 | 89.4 | 31.1 | **99.9** |
| | WP-s | 96.4 | 78.4 | 92.9 | 87.9 | 93.2 | 95.4 | 69.3 | 38.4 | **99.5** |

In Table 11, we report the LGT detection performance of ReMoDetect and other detection baselines by considering additional performance metrics, including true positive rate (TPR) at 1% false positive rate (FPR) and Area under the precision-recall curve (AUPR). As shown in the table, ReMoDetect significantly and consistently outperforms in all cases.

## B.4 Dataset Examples and Predicted Reward Scores

Table 12: Dataset examples with their model, domain, and ReMoDetect score.

| Model | Domain | Example | ReMo-D Score |
|-------|--------|---------|--------------|
| Human | WP-s | You are a teenager with the ability to measure how "Dangerous" people are on a scale from 1 to 10 just by looking at them. A normal child would be a 1, while a trained man with an assault rifle might be a 7. Today, you notice the unassuming new kid at school measures a 10. When I see people, I see numbers. Just floating there. I can't remember since when it's begun, but that's not important. An innocent baby would measure as 1. A fellow high school student, typically a 3 or 4 - there's a lot of stress involved. The day before my best friend stabbed his entire family to death and hung himself, he was abnormally quiet - a 9. Just seeing a living being's image is enough: on television, the rioters at Ferguson measured around 6 and 7. When I was daring enough to watch videos of ISIS members committing atrocious murders: a 9. That day | -6.01 |
| GPT4 Turbo | WP-s | You are a teenager with the ability to measure how "Dangerous" people are on a scale from 1 to 10 just by looking at them. This unusual power has made you both cautious and curious about the people you encounter in your daily life. One day, while walking through a busy market, you notice a man with a danger level of 9. Intrigued, you decide to follow him discreetly, keeping a safe distance. As you observe his interactions, you notice he has a commanding presence and seems to be involved in some shady dealings. However, as you continue to watch, you see him also display moments of kindness, helping an elderly woman with her groceries and playing with a group of children. This experience teaches you that people are complex and multi-dimensional, and that danger levels don't necessarily define someone's entire character. You learn to use your power with more discernment, understanding that everyone has both light and dark sides. | 4.91 |
| Human | XSum | Regina Hungerford, 56, denied striking a 17-year-old as he listened to music in her class at Merthyr Tydfil College. But a disciplinary panel ruled contact with the pupil was probably made and her conduct had been "unacceptable". On Thursday, the Education Workforce Council imposed the suspension, saying: "The public interest is in favour of her being able to teach again." Mrs Hungerford admitted shouting and slamming a book on the desk of the "provocative and disruptive" teenager as he listened to rap music in her classroom - but always denied hitting his head. She was cleared, on appeal, of a criminal charge for assaulting the pupil during a maths lesson for those with learning difficulties. But in November, a disciplinary panel found, on the balance of probabilities, she had made physical conduct with the pupil on the head or hand. The panel found she had | -1.70 |
| Claude3 Opus | XSum | Regina Hungerford, 56, denied a 17-year-old while she listened to music in her class at Merthyr Tydfil College. The incident occurred when the student refused to take his headphones off during a lesson. Hungerford, who has taught at the college for more than a decade, maintained her innocence throughout the investigation. Witnesses claim that the altercation began when Hungerford approached the student and requested that he stop listening to music and focus on the lesson. The student apparently ignored her request, resulting in a heated exchange. Several classmates reported seeing Hungerford striking the student while others stated that they did not witness physical contact. The college administration has launched an internal investigation into the matter and Hungerford has been suspended pending the outcome. The student's family has been notified and the local authorities are also examining the incident. The college has declined to comment on the matter. | 4.04 |

In Table 12, we show dataset examples and their ReMoDetect score.

## B.5 Additional Observational Studies

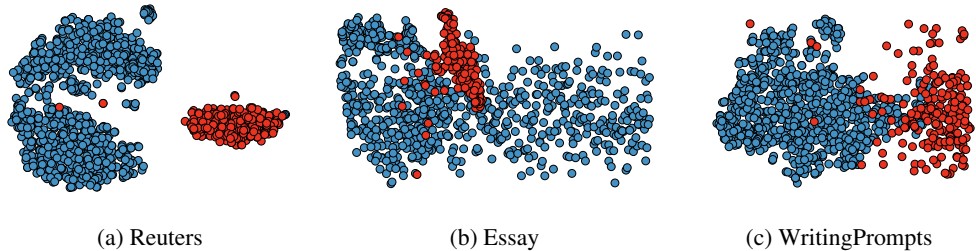

(a) Reuters        (b) Essay        (c) WritingPrompts

Figure 7: t-SNE of the reward model's final feature in multiple domains Reuters, Essay, Writing-Prompts generated by GPT3.5/GPT4 Turbo, Llama3-70B-instruct, and Claude3 Opus.

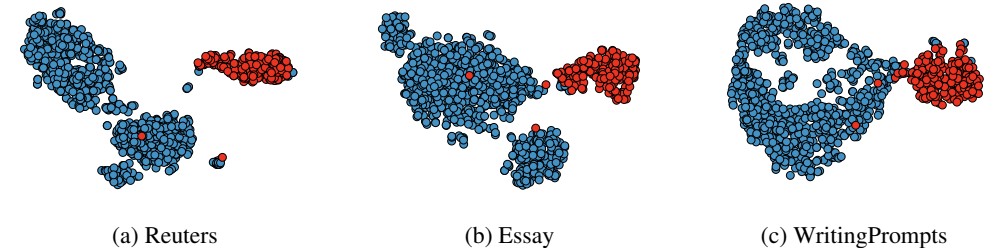

(a) Reuters        (b) Essay        (c) WritingPrompts

Figure 8: t-SNE of the ReMoDetect's final feature in multiple domains Reuters, Essay, Writing-Prompts which generated by GPT3.5/GPT4 Turbo, Llama3-70B-instruct, and Claude3 Opus

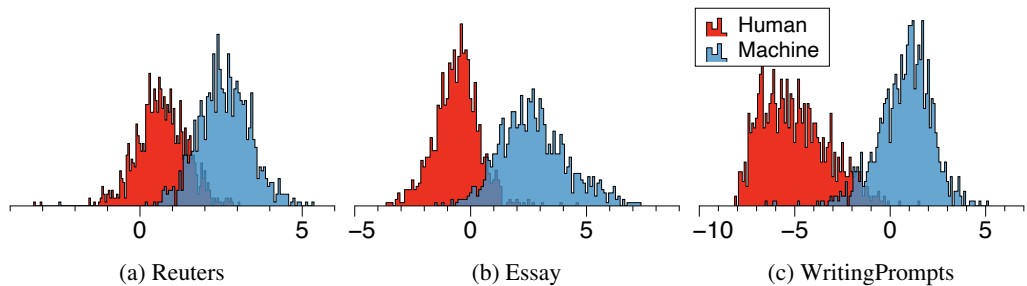

(a) Reuters        (b) Essay        (c) WritingPrompts

Figure 9: Reward distribution of the reward model in multiple domains Reuters, Essay, Writing-Prompts generated by GPT4 Turbo, and Claude3 Opus.

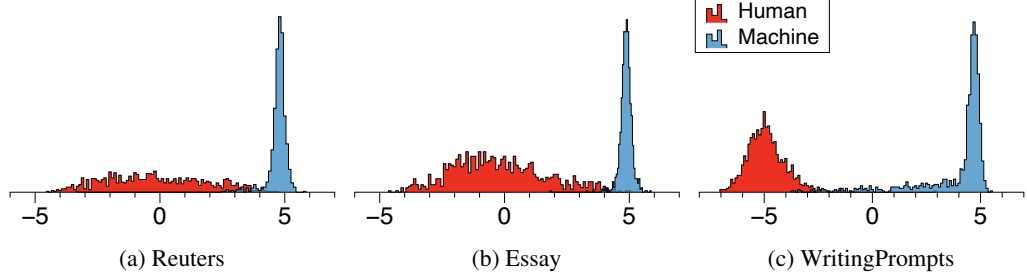

(a) Reuters        (b) Essay        (c) WritingPrompts

Figure 10: Reward distribution of the ReMoDetect in multiple domains Reuters, Essay, Writing-Prompts which generated by GPT4 Turbo, and Claude3 Opus.

In Figure 7, and Figure 8, we present t-SNE of the reward model and ReMoDetect. Figure 9 and Figure 10 display the reward distribution. These figures demonstrate that, even without further training, the reward model can distinguish between human-written texts and LGT. Additionally, ReMoDetect emphasizes the separation between human-written text and LGT.

## B.6 Robustness of Reward Models against Rephrasing Attacks

Table 13: Robustness against rephrasing attacks. We report the average AUROC (%) before ('Original') and after ('Attacked') the rephrasing attack with T5-3B on the Fast-DetectGPT benchmark, including XSum, PubMed, and WritingPrompts-small. Values in the parenthesis indicate the relative performance drop after the rephrasing attack. The bold indicates the best result.

| Model | Accuracy | Loglik. | D-GPT | NPR | FD-GPT | Ours (reward model) | Ours (ReMoDetect) |
|---|---|---|---|---|---|---|---|
| GPT4 | Original | 82.1 | 69.0 | 68.1 | 90.6 | 79.0 | **97.9** |
| | Attacked | 63.7 (-22.4%) | 44.8 (-35.1%) | 47.0 (-31%) | 74.5 (-17.7%) | 71.2 **(-9.9%)** | **87.2** (-10.9%) |
| Llama3 70B | Original | 93.5 | 84.9 | 84.9 | 96.8 | 80.9 | **98.5** |
| | Attacked | 79.9 (-14.5%) | 61.7 (-27.4%) | 64.7 (-23.7%) | 87.9 **(-9.2%)** | 71 (-12.3%) | **88.3** (-10.4%) |
| Gemini pro | Original | 79.2 | 71.3 | 75.8 | 79.9 | 64.1 | **81.8** |
| | Attacked | 64.9 (-18%) | 50.7 (-28.9%) | 55.7 (-26.6%) | 64.5 (-19.3%) | 55.8 **(-13%)** | **67.4** (-17.6%) |

In Table 13, we compare the robustness against the paraphrased attack of the reward model and other baselines including ReMoDetect. The experiment shows that the reward model is robust against paraphrasing attacks (i.e. reward model and ReMoDetect are the two least drops against paraphrasing attacks). From the results, we hypothesize that the robustness against attack came from the reward model itself. Conceptually the human preference for the text samples doesn't change much as the distribution shifts or paraphrases some words, hence, the reward score is independent of the minor variation of the sentence. We believe that the result of the experiment supports our hypothesis. Furthermore, exploring the characteristics and applications of the reward model would be interesting in the future.

## B.7 Additional ReMoDetect Models Trained From Differently Initialized Reward Models.

Table 14: Comparison of multiple ReMoDetect models trained from reward models, including deberta, Gemma-2B (G. 2B), Llama3-8B (L. 8B). We report the average AUROC (%) on the fastdetectGPT benchmark, including PubMed, XSum, and WritingPrompts-small (WP-s).

| Model | Domain | FD-GPT | Open-D | Ours (deberta) | Ours (G. 2B) | Ours (L. 8B) |
|---|---|---|---|---|---|---|
| GPT3.5 Turbo | PubMed | 90.2 | 61.9 | **96.4** | 90.1 | 94.7 |
| | XSum | 99.1 | 91.5 | 99.8 | **100.0** | **100.0** |
| | WP-s | 99.2 | 70.9 | **99.9** | **99.9** | 99.7 |
| GPT4 | PubMed | 85.0 | 53.1 | **96.1** | 91.4 | 92.1 |
| | XSum | 90.7 | 67.8 | 98.8 | 99.9 | **100.0** |
| | WP-s | 96.1 | 50.7 | 98.7 | **99.6** | 99.4 |
| GPT4 Turbo | PubMed | 88.8 | 55.8 | **97.0** | 91.2 | 92.9 |
| | XSum | 97.4 | 88.2 | 99.8 | **100.0** | **100.0** |
| | WP-s | 99.4 | 72.3 | **100.0** | **100.0** | **100.0** |
| Llama3 70B | PubMed | 90.8 | 52.9 | **96.3** | 91.8 | 94.3 |
| | XSum | 99.7 | 96.2 | 99.5 | **100.0** | 99.9 |
| | WP-s | **99.9** | 77.5 | 99.8 | 99.6 | 99.6 |
| Gemini pro | PubMed | 82.1 | 57.3 | **85.6** | 78.8 | 81.8 |
| | XSum | 79.5 | 72.2 | **88.2** | 87.5 | 85.3 |
| | WP-s | 78.0 | 70.2 | 71.6 | 84.2 | **89.2** |
| Calude3 Opus | PubMed | 88.2 | 48.9 | **96.4** | 90.9 | 93.3 |
| | XSum | 96.2 | 86.2 | 99.5 | **99.9** | 99.8 |
| | WP-s | 93.5 | 65.7 | **99.9** | 99.7 | 99.8 |
| Average | - | | 91.9 | 68.9 | **95.8** | 94.6 | 95.6 |

We additionally consider the ReMoDetect models trained from differently initialized reward models. To address the consideration, we conducted experiments to train ReMoDetect using three reward models. As shown in Table 14, ReMoDetect models consistently outperform other baselines, even though the model trained from differently initialized reward models. Nonetheless, the ReMoDetect's detection performance can vary with initialization. Thus, we suggest interesting future works to find a better detector, such as ensembling several trained models or using an enhanced reward model.

### B.8 Comparison with GPTZero per domain

Table 15: Detection Score of GPTZero [39], a commercial black-box LGT detection API. We report the AUROC (%) on the Fast-DetectGPT benchmark, including PubMed, XSum, and WritingPrompts.

| Model | Domain | GPT 3.5 Turbo | GPT4 | GPT4 Turbo | Llama3 70B | Gemini pro | Claude3 Opus |
|---|---|---|---|---|---|---|---|
| GPTZero | PubMed | 88.0 | 84.8 | 87.2 | 90.1 | 83.2 | 88.0 |
| | XSum | 99.5 | 98.2 | 100.0 | 100.0 | 85.8 | 99.9 |
| | WP-s | 92.9 | 82.6 | 100.0 | 99.8 | 79.7 | 99.1 |
| ReMoDetect | PubMed | 96.4 | 96.1 | 97.0 | 96.3 | 86.4 | 96.4 |
| | XSum | 99.8 | 98.8 | 99.8 | 99.5 | 74.5 | 99.5 |
| | WP-s | 99.9 | 98.7 | 100.0 | 99.8 | 86.4 | 99.9 |

In Table 15, we report the performance of GPTZero [39] and ReMoDetect in PubMed, XSum, and WritingPrompts (note that Table 3 reports the average AUROC of these domains). It is worth noting that ReMoDetect outperforms in most of the cases and consistently shows better performance in PubMed (which is an expert domain), indicating the effectiveness ReMoDetect on low-data regimes.

### B.9 Comparison on Aligned Small Language Models

Table 16: AUROC (%) of multiple LGT detection methods, including log-likelihood (Loglik.), Rank, Fast-DetectGPT (FD-GPT), OpenAI-Detector (Open-D), ChatGPT-Detector (Chat-D), and ReMoDetect (Ours). We consider LGT detection benchmarks from Fast-DetectGPT: PubMed, XSum, and WritingPrompts-small(WP-s). The bold indicates the best result within the group.

| Model | Domain | Loglik. | Rank | FD-GPT | Open-D | Chat-D | **Ours** |
|---|---|---|---|---|---|---|---|
| Llama3 8B-it | PubMed | 85.0 | 60.4 | 89.6 | 53.7 | 33.4 | **94.6** |
| | XSum | 82.3 | 68.9 | 86.8 | **95.4** | 13.1 | 85.4 |
| | WP-s | 87.2 | 72.3 | 91.0 | 81.2 | 26.4 | **95.5** |
| Gemma2 9B-it | PubMed | 69.8 | 55.9 | 71.6 | 36.4 | 85.1 | **95.1** |
| | XSum | 85.1 | 69.4 | 94.0 | 74.0 | 97.7 | **99.5** |
| | WP-s | 86.7 | 71.9 | 96.6 | 50.1 | 70.3 | **96.8** |
| Gemma2 2B-it | PubMed | 67.9 | 56.6 | 72.3 | 44.4 | 78.1 | **90.0** |
| | XSum | 82.1 | 18.2 | 89.8 | 67.6 | 97.2 | **94.9** |
| | WP-s | 84.6 | 71.8 | **99.0** | 70.8 | 63.7 | 94.2 |
| Qwen2 1.5B-it | PubMed | 82.3 | 61.0 | 89.8 | 62.9 | 23.9 | **92.7** |
| | XSum | 96.5 | 66.7 | 98.3 | 97.2 | 1.3 | **99.6** |
| | WP-s | 97.5 | 78.2 | 98.6 | 94.3 | 17.7 | **99.1** |
| OLMo 7B-sft | PubMed | 88.4 | 60.5 | 92.8 | 62.0 | 23.6 | **94.1** |
| | XSum | 96.6 | 66.0 | **99.1** | 97.3 | 5.9 | 98.1 |
| | WP-s | 98.1 | 78.5 | 98.8 | 95.2 | 19.5 | **99.2** |
| Average | - | 86.0 | 63.8 | 91.2 | 72.2 | 43.8 | **95.3** |

We additionally consider small aligned models particularly when the model parameter size is smaller than 10B, including Llama3-8b, Gemma-2-9b, Gemma-2-2b, Qwen2-1.5b-it, and Olmo7b-sft. As shown in Table 16, ReMoDetect also effectively detects LGT of small language models. For instance, ReMoDetect achieves 97.1% average AUROC in Qwen2-1.5b-it while the second-best reaches 84.8%.

