# OpenReview forum: "ReMoDetect: Reward Models Recognize Aligned LLM's Generations"
_NeurIPS.cc/2024/Conference — NeurIPS 2024 poster_

### Official Review · Reviewer_3CSf · 2024-07-10

**Soundness:** 3
**Presentation:** 3
**Contribution:** 3
**Rating:** 5
**Confidence:** 3

**Summary:**

The authors demonstrate that reward models inherently possess the capability to distinguish between human-written and machine-generated text. They propose a method for continuous pairwise fine-tuning of existing RMs, which achieves excellent results on several LLM-generated text (LGT) detection datasets and exhibits robustness against adversarial attacks.

**Strengths:**

1. The premise of the study is interesting. Given that many LLMs are optimized based on RMs, it stands to reason that RMs encode certain features of LLM-generated text, making them a promising foundation for further training in LGT detection.
2. The experimental results presented by the authors are impressive.

**Weaknesses:**

1. While the performance improvements are noteworthy, further clarification is needed regarding the mechanisms behind these improvements. For instance, how does the pairwise loss function proposed in this paper differ from loss functions used in other LGT detection methods?
2. The authors' experiments validate the accuracy of only one RM after continuous training using their proposed method, which is the same as in the preliminary experiments. However, the scope claimed in the paper seems to encompass all RMs. Either additional experiments should be conducted or the claim should be narrowed.
3. Although utilizing RMs as a starting point is an interesting approach, the availability of RMs transforms what would be a black-box detection into a semi-white-box detection. This difference in setup may naturally lead to performance improvements, which should be addressed.
4. If a model only undergoes the supervised fine-tuning phase of alignment, can the proposed LGT model still be effective?

**Questions:**

1. In Table 2, can the authors report the parameter counts for the baseline methods in the main experiments?
2. In Table 2, have smaller models been tested, rather than focusing solely on large models?
3. Regarding Table 4 and 5, please provide convincing explanations for the enhanced performance against distribution shifts and attacks.
4. In Figure 5, the score distribution for human-written text shows increased variance after training, while the opposite is true for LLM-generated text. Can the authors provide an explanation for this phenomenon from the perspective of the loss function?

**Limitations:**

We encourage the authors to provide a separate section for limitations and social impacts.

---

> ### Author Rebuttal · Authors · 2024-08-07
>
> Dear reviewer 3CSf,\
> We sincerely appreciate your efforts and comments to improve the manuscript. We respond to your comment in what follows.
>
> ---
>
> **[W1] Further clarification regarding the mechanisms behind the improvements is needed.**
>
> We clarify that ReMoDetect is effective due to the following reasons. First, the reward model itself is already effective at detecting the LLM-generated texts (LGTs) as the reward model gives a higher preference to the LGTs compared to human-written texts. Second, we have designed objectives that encourage the model to prefer LGTs even further to increase the detection performance. Note that the pair-wise loss of continual preference tuning maximizes the predicted preference gap between LGTs and human written texts. Furthermore, note that our objective is quite new to the LGT detection community, as recent effective detection methods [1,2,3] mostly focus on inference time detection scores, e.g., measuring probability change when the input text is perturbed.
>
> [1] Detectgpt, ICML 2023.\
> [2] Fast-detectgpt, ICLR 2024.\
> [3] Detectllm, arXiv 2023.
>
> ---
>
> **[W2] Experimented on only one RM, the claim of the paper scope should be narrowed.**
>
> Thank you for pointing this out. To address your concern, we conducted additional experiments using three reward models: Deberta 500M,  Gemma 2B, and Llama3 8B based RM. As shown in Table 4 in the attached pdf, all reward models trained with ReMoDetect consistently outperform other baselines, indicating that the reward model is indeed effective for detecting aligned LGTs. We thank the reviewer for the suggestion and will incorporate the result in the final draft.
>
> ---
>
> **[W3] Availability of RM transforms black-box detection into a semi-white-box detection**
>
> We respectfully yet strongly disagree that the availability of RM transforms our method from black-box to semi-white-box detection. Note that there exist multiple open-source RMs [1,2,3] where all results are based on publicly available RMs, including the new experiments in [W2]. Furthermore, we have tested all text detection without access of the LLM’s information, making the experimental setup fully black-box.
>
> [1] OpenAssistant/reward-model-deberta-v3-large-v2\
> [2] weqweasdas/RM-Gemma-2B\
> [3] sfairXC/FsfairX-LLaMA3-RM-v0.1
>
> ---
>
> **[W4] Is ReModetect still effective in detecting SFT only models?**
>
> Thank you for an interesting question. To examine whether ReMoDetect is also effective in detecting SFT only model (i.e., no RLHF), we have additionally conducted an experiment on Olmo7b-sft [1] model.  As shown in Table 3 in the attached pdf, ReMoDetect effectively detects LGT from SFT only model, e.g. Olmo7b-sft: 97.1%  in average AUROC (%) while the second best achieves 91.2 (%). We believe this is because the SFT implicitly trains the model to reflect the human preference from the instruction tuning dataset [2], thus making the ReMoDetect well-detect the texts from SFT models.
>
> [1] allenai/OLMo-7B-SFT-hf\
> [2] Self-Play Fine-Tuning Converts Weak Language Models to Strong Language Models, ICML 2024
>
> ---
>
> **[Q1] Report the parameter counts for the baseline methods in Table 2**
>
> Thank you for pointing this out. We have reported the parameter count of the baseline method and ours in Table 10, Appendix B.4. For your convenience, we have also reported in Table 2 in the attached pdf. Here, as DetectGPT, NPR, and Fast-DetectGPT used two models (i.e., a base model and a perturbing model), we have separately reported the numbers. As shown in the table, our method has significantly fewer parameters compared to the second best (i.e., Fast-DetectGPT) yet outperforms the baseline.
>
> ---
>
> **[Q2] In Table 2, have smaller models been tested?**
>
> First, we carefully remark that we already considered small models, including the Phi-3 series (i.e. each has sizes of 3.8B, 7B, and 14B) where ReMoDetect consistently outperforms other methods on those small models (in Appendix B.3). Nevertheless, to further address your question, we additionally experimented with other small models, including Llama3-8b, Gemma2-9b, Gemma2-2b, Qwen2-1.5b-it, and Olmo7b-sft. As shown in Table 3 in the attached pdf, ReMoDetect also effectively detects LGT of small models. For instance, ReMoDetect achieves 97.1% of average AUROC in Qwen2-1.5 while the second-best reaches 84.8%.
>
> ---
>
> **[Q3] Why does the method show enhanced performance against distribution shifts and attacks?**
>
> Thank you for your constructive questions. We believe that robustness against distribution shifts and attacks came from the reward model itself.
> Conceptually, the human preference (or the quality) of the text samples doesn’t change much as the distribution shifts or paraphrases some words, hence, the reward score is independent from the minor variation of the sentence. Additionally, we conducted experiments to test our conceptual hypothesis. As shown in Table 1, the reward model is robust against paraphrasing attacks (i.e. RM and ReMoDetect are the two least drops against paraphrasing attacks). We believe that the result of the additional experiment supports our hypothesis. Furthermore, exploring the characteristics and applications of the reward model would be interesting in the future.
>
> ---
>
> **[Q4] Why does the score distribution for human-written texts show increased variance after training, while the opposite is false for LGTs?.**
>
> It is true that our objective focuses on increasing the preference gap between LGTs and human-written texts, where having high variance in human-written texts are not explicitly defined in the objective. We conjecture this phenomenon occurred as the quality in the human written text varies (as multiple individuals across various backgrounds have written the text) while aligned LLMs share somewhat similar training recipes across models, leading to more consistent output patterns.
>
> ---
>
> **[L1] Separate Social Impact and Limitation**
>
> In the final draft, we will separate social impact and limitations.

---

### Official Review · Reviewer_dzQj · 2024-07-10

**Soundness:** 3
**Presentation:** 3
**Contribution:** 3
**Rating:** 7
**Confidence:** 4

**Summary:**

This paper proposes a method named ReMoDetect to use a reward model for model-generated text detection. Firstly, The authors find that the existing reward model can easily distinguish human-written text from language-model-generated responses. Then, the authors propose two techniques, 1) continual preference training and 2) mixed human and LLM responses, to further train the reward model for LLM-generated text detection. Experimental results demonstrate the effectiveness of the proposed reward model based LLM-generated text detection.

**Strengths:**

- The motivation is clear, and the proposed method to use a reward model for detection is original.
- Based on the experimental results, the proposed method is effective across multiple LLMs and different domains.
- The paper is well-organized and easy to follow.

**Weaknesses:**

- To evaluate a response using a reward model, it requires both the given context or prompt x and the response y. However, the prompt x is not always available in the LLM-generated text (LGT) detection problem. It would be useful to illustrate how the corresponding prompts are determined when evaluating the proposed models. And it would be great to additionally evaluate the proposed method on datasets without prompts.
- It is unclear why the reward model can recognize the LLM-generated texts from human-written responses.

**Questions:**

- Is there some correlation between the reward model accuracy and the corresponding LGT detection accuracy?

---

> ### Author Rebuttal · Authors · 2024-08-07
>
> Dear reviewer dzQj,
>
> We sincerely appreciate your efforts and comments to improve the manuscript. We respond to your comment in what follows.
>
> ---
>
> **[W1] How the corresponding prompts are determined when evaluating the proposed models. RM works for the prompt $x$ given. What about prompt ungiven cases?**
>
> While the initial context $x$ and the generation $y$ (by LLM or human) is explicitly defined in the training dataset and objective, the RM only observes the concatenation of $x$ and $y$ (i.e., the full paragraph) to predict the reward score. Therefore at test-time, we also give the full paragraph to the RM without any indication of initial context and generated part as in the training setup.
>
> ---
>
> **[W2] Unclear why RM can recognize LGT from human-written responses**
>
> We believe it is because of the alignment training objective that recent LLMs have utilized. Note that alignment training make the LLM to generate texts with high predicted rewards (i.e., human preference), thereby well trained LLMs are likely to generate text with higher rewards compared to human. This is analogous to the phenomenon that a Go model optimized to maximize the reward (i.e. winning the game) frequently surpasses human experts in the game.
>
> -------
>
> **[Q1] Correlation between the RM accuracy and the corresponding LGT detection accuracy?**
>
> Thank you for an interesting question. To this end, we have considered three reward models (RMs), namely, DeBerta 500M, Gemma 2B, and Llama3 8B based RM, where the reward accuracies are 61.8, 63.9, and 84.7, respectively (measured in RewardBench [1]). As shown in Table 4 in the attached pdf, we found some interesting correlations where larger models actually perform better than smaller models on long context detection. For instance, DeBerta, Gemma, and Llama3 based ReMoDetect achieved an average AUROC over two long context datasets (i.e., WritingPrompt-S, XSum) of 96.3, 97.5, 97.6, respectively. We observed that DeBerta outperforms other models in short context dataset (i.e., PubMed), possibly because DeBerta has trained on short context for pre-training (context size of 512). We thank the reviewer for the question and will incorporate the result in the final draft.
>
> [1] https://huggingface.co/spaces/allenai/reward-bench

---

> > ### Comment · Reviewer_dzQj · 2024-08-13
> >
> > Thanks for the response! I have read all reviews and the corresponding author responses. These comments are helpful and address some of my concerns, which can improve the quality of the manuscript if included in the revision. However, I still
> > confused about the W1, how cases without the prompts, in which only $y$ can be observed, to work with this method. So I keep my score for now.

---

> > > ### Author Response · Authors · 2024-08-13
> > > **Thank you for the response.**
> > >
> > > Dear reviewer dzQj,
> > >
> > > We sincerely thank the reviewer for the response and the effort in reading our response. We would like to respond to the remaining concern about [W1].
> > >
> > > ---
> > > **[W1]  How cases without the prompts, in which only $y$ can be observed, to work with this method.**
> > >
> > > We want to clarify again that our method doesn't need to see prompt $x$. While the input of RM is a concatenation of $x$ and $y$, ReMoDetect only observes $y$ as shown in the example below. We tested all evaluations in our paper by only using $y$ without $x$.
> > >
> > >
> > > **Input of original RM: $x + y$**
> > > ```
> > > "Please write an article with 500 words. A man forgets to water his potted plant for a whole week …"
> > > ```
> > >
> > > **Input of ReMoDetect: $y$**
> > > ```
> > > "A man forgets to water his potted plant for a whole week …"
> > > ```

---

> > > > ### Comment · Reviewer_dzQj · 2024-08-13
> > > >
> > > > Thanks for the detailed illustration. My concerns are addressed and I would like to raise my score from 6 to 7.

---

> > > > > ### Author Response · Authors · 2024-08-13
> > > > > **Thank you for your response.**
> > > > >
> > > > > Dear reviewer dzQj,
> > > > >
> > > > > We are very happy to hear that our rebuttal addressed your concerns well. \
> > > > > Due to your valuable and constructive suggestions, we do believe that our paper is much improved.
> > > > >
> > > > > If you have any further questions or suggestions, please do not hesitate to let us know.
> > > > >
> > > > > Thank you very much,  \
> > > > > Authors

---

### Official Review · Reviewer_kLmr · 2024-07-13

**Soundness:** 3
**Presentation:** 4
**Contribution:** 4
**Rating:** 7
**Confidence:** 4

**Summary:**

The paper is about a novel and effective approach for LLM-generated text (LGT) detection by making use of the reward model score. The authors observe that LGT often has higher reward model score compared to human-written texts. They then further increase the separating by fine-tuning the reward model to score LGT higher than human-written texts, and use additional LLM-rephrased human-written texts as the median preference text to assist with the learning.

**Strengths:**

- The evaluation results are very strong on selected benchmarks, outperforming other LGT detection methods.
- The analysis about robustness on unseen distributions, rephrasing attacks and input response length is solid.

**Weaknesses:**

- The work lack qualitative analysis on examples. The cases and patterns for errors, and for improvements, are unclear.

**Questions:**

- Why GPTZero is not evaluated on MGTBench?
- While the Table 2 and 3 covers 6 models, why Table 4 only covers 4 models, Figure 4 only covers a combination of 2 models, and Figure 5 only covers 1 model?

**Limitations:**

- The paper has inconsistency in selecting the results to report in both the main paper and the appendix. See questions for details.
- The selected benchmarks are mostly scientific writing and news writing, while other commonly used benchmarks in Question Answering, Web Text and Story Generation (as defined in [1]) are not covered.

[1] Wu, J., Yang, S., Zhan, R., Yuan, Y., Wong, D. F., & Chao, L. S. (2023). A survey on llm-gernerated text detection: Necessity, methods, and future directions. arXiv preprint arXiv:2310.14724.

---

> ### Author Rebuttal · Authors · 2024-08-07
>
> Dear reviewer KLmr,
>
> We sincerely appreciate your efforts and comments to improve the manuscript. We respond to your comment in what follows.
>
> ---
> **[W1] Lack of qualitative analysis on examples**
>
> Thank you for your constructive comment. While we have a portion of qualitative examples in Appendix B.5, we will add more comprehensive qualitative analysis in the final draft.
>
> First, existing detection methods struggle to detect short passages, as shown in Figure 6 in our paper. Nevertheless, ReModetect outperforms other baselines for shorter-length passages, and there can be more improvement points for future works.
>
> Second, human-written text tends to have more grammar errors compared to LGT as shown in the examples below. To verify the observation, we compared the number of grammar errors in the samples from the WritingPrompts-small dataset by using Grammarly [1]. The average error per 100 words for LGT of GPT4 Turbo was 0.486, while for human-written text was 2.278. We believe this observation suggests that ReMoDetect can serve as a text quality detector.
>
>
> **Human - grammar error: 11, ReMoDetect Score: - 5.35**
> ```
> A man forgets to water his potted plant for a whole week. This negligence starts a long chain reaction that leads up to World War III. My plant had died. Because my plant died I went to the store to get a new one. On the way there I cut a man off in traffic. The light changed and I was in the right of way thats the important part. The man I cut off was an ambassador to Russia, back in North America for a quick visit with other officials. He was on the phone to a Russian Delegate he was making peace treaties with should escalations ever occur. Because he was cut off he began to scream obscenities into the phone. Before he could explain the Russian Delegate had yelled back and hung up the phone. Before he could call back the Russian had contacted other Delegates to begin non aggressive hostilities. This rise
> ```
>
> **GPT4 Turbo  - grammar error: 2, ReMoDetect Score: 4.87**
>
> ```
> A man forgets to water his potted plant for a whole week. This negligence starts a long chain reaction that leads up to World War III. The plant, a rare species, is the last of its kind and a crucial ingredient in a serum that can cure a deadly virus. Scientists from around the world are counting on the plant's survival to mass-produce the antidote. When the man finally remembers to water the plant, it's too late. The plant has withered and died. News of the plant's demise spreads quickly, causing panic and fear. Countries begin to blame each other for not doing enough to protect the plant and secure the cure. Tensions rise, alliances are broken, and diplomatic relations deteriorate. Eventually, conflicts erupt, and the world is plunged into
> ```
> [1] Grammarly: https://app.grammarly.com/
>
> ---
>
> **[Q1] GPTZero on MGTBench.**
>
> Thank you for pointing this out. During the initial development, we actually compared ReMoDetect with GPTZero on MGTBench, but omitted the result as both methods showed near 100\% of detection score, making it hard to extract meaningful conclusions from the result (see the table below). We believe making more hard benchmarks will be an interesting future direction to explore.
>
> \begin{array}{lcccc}\hline
> \text{Model} & \text{GPT3.5 Turbo} & \text{GPT4 Turbo} & \text{Llama3 70B} & \text{Gemini pro} \\ \newline\hline
> \text{GPTZero} & 100.0 & 99.9 & 99.9 & 100.0 \\ \newline
> \text{Ours} & 100.0 & 99.9 & 99.9 & 99.9 \\ \newline\hline
> \end{array}
>
> ---
>
> **[Q2 & L1] Inconsistent model on Figure4,5 Table2,3,4.**
>
> In the main paper, we tried to compare all aligned LLMs in the main experiments (including Table 2,3) while only considering the most recent powerful LLMs (e.g., GPT4 and Claude 3 Opus) for the analysis. Nevertheless, we agree with the reviewer that this can be seen as an inconsistency for selection. To this end, we have conducted additional experiments on more aligned LLMs: GPT3.5 Turbo, GPT4, GPT4 Turbo, Llama3 70B, Gemini Pro, Claude Opus as shown in Figure 1, Figure 2, and Table 1 in the attached pdf.
>
> ---
>
> **[L2] Benchmarks are mostly scientific writing and news writing. Benchmarks in QA, Webtext, and Story Generation are not covered.**
>
> We clarify that we have already covered QA and story generation. Note that according to the reference the reviewer mentioned [1], PubMed is QA, and WP is story generation.
>
> [1] A survey on llm-gernerated text detection: Necessity, methods, and future directions, arXiv 2023

---

### Official Review · Reviewer_CEwj · 2024-07-19

**Soundness:** 3
**Presentation:** 3
**Contribution:** 3
**Rating:** 7
**Confidence:** 3

**Summary:**

The paper finds that reward models used in RLHF can detect texts generated by LLMs. Based on this, the paper presents ReMoDetect, a novel method that further trains the reward model using continual preference fine-tuning and a challenging text corpus rephrased by LLMs. ReMoDetect achieves new SOTA on various LGT benchmarks.

**Strengths:**

1. The paper presents an interesting finding that RLHF reward models can make LLMs generate outputs that align too closely with human preferences, even more so than human-written texts. This motivates the authors to further fine-tune the reward model, which is a well-motivated approach.
2. The authors conduct extensive experiments showing that the proposed method achieves SOTA on various benchmarks. The ablation study demonstrates the effectiveness of each component in ReMoDetect.
3. The paper is well-written and easy to follow, with a clear storyline from motivation to proposed methodologies. The two training strategies are simple and reasonable.
4. Detecting LGT is an important research problem.

**Weaknesses:**

1. The proposed method relies on the quality of the reward models, such as the training dataset and model parameters. A pre-trained reward model may be biased towards a specific dataset and may not generalize well to all LLMs. Poor initialization of the reward model may harm performance.
2. Reward models are also LLMs. Using such models for LGT detection involves long inference times.

**Questions:**

1. What is the performance difference when using different reward models for initialization? For example, if we use the reward model of LLM A, and then classify the results of both LLM A and LLM B, does the LGT classifier perform better at detecting LGT of LLM A?
2. If the reward model of a specific LLM is not available and the reward model has learned some specific or undesirable preferences, how does the proposed method perform in this scenario?

**Limitations:**

The reward models are not available for some closed-source LLMs.

---

> ### Author Rebuttal · Authors · 2024-08-07
>
> Dear reviewer CEwj,
>
> We sincerely appreciate your efforts and comments to improve the manuscript. We respond to your comment in what follows.
>
> ---
>
> **[W1] The method relies on the quality and initialization of the RM.**
>
> First, we would like to clarify that we have only trained a single reward model for ReMoDetect, which is used across all experiments (i.e., we did not train separate ReMoDetect for individual datasets or aligned LLMs).
>
> Nevertheless, we follow the suggestion to show that the method does not rely on an initialized reward model. To address your concern, we conducted experiments to train ReMoDetect using three reward models. As shown in Table 4 in the attached pdf, ReMoDetect models consistently outperform other baselines, even though the model trained from differently initialized reward models. Nonetheless, the ReMoDetect’s detection performance can vary with initialization. Thus, we suggest interesting future works to find a better detector, such as ensembling several trained models or using an enhanced reward model.
>
> ---
>
> **[W2] ReMoDetect may involve long inference time.**
>
> We remark that ReMoDetect is highly efficient in terms of inference time and memory compared to other LGT detection methods (in Appendix B.4). As shown in Table10 in our paper, ReMoDetect is 7.2 times faster and uses a 17.4 times smaller model than the second-best model, Fast-DetectGPT.
> This is because the recent LGT detection methods require multiple forward of detector LLM to estimate the score (e.g., DetectGPT, NPR, and fast-DetectGPT perturb the text multiple times to capture the probabilistic curvature) while our method only requires a single forward pass to compute the score.
>
> ----
>
> **[Q1] Does a ReMoDetect trained from LLM A perform better at detecting LGT of LLM A than other LLMs? LGT classifier detect outputs better from a specific LLM when initialized with that LLM's reward model?**
>
> We couldn’t verify whether the use of the reward model of LLM A classifies better at detecting LGT of LLM A than LGT of LLM B because we cannot access the reward model of large models (as even opensource models do not opensource the reward models).  However, it is worth noting that we have demonstrated that a single ReMoDetect model can effectively detect LGT in many different LLMs, showing the detection generalization. We also agree it will be an interesting experiment if the RMs of aligned LLMs (e.g., RM used for training Gemini) get open-sourced.
>
> ---
> **[Q2 & L1] How does ReMoDetect perform if the reward model of a specific LLM is not available?**
>
> As we clarified in the response of [W1], we have only trained an open-source reward model for ReMoDetect, which is used across all experiments (i.e., we did not train separate ReMoDetect for individual datasets or aligned LLMs). From the results, we believe ReMoDetect don’t need access to RM of closed LLM.

---

### Author Rebuttal · Authors · 2024-08-07

Dear reviewers and AC,

We sincerely appreciate your valuable time and effort spent reviewing our manuscript. As reviewers highlighted, we believe our paper tackles an interesting and important problem (CEwj) and provides an effective (all reviewers) framework for detecting LGT, which is motivated by interesting findings (CEwj,  dzQj), validated with extensive evaluations(CEwj, dzQj) followed by a clear presentation (all reviewers).

We appreciate your constructive comments on our manuscript. In the attached pdf, we have run the following additional experiments to clarify the reviewer's comments:

- Detection Accuracy of ReMoDetect under paraphrased attack (Table 1)
- Model Parameter of ReMoDetect and baselines (Table 2)
- Detection Accuracy of ReMoDetect on small LMs (Table 3)
- Comparison of ReMoDetect models initialized from several reward models (Table 4)
- Reward Score distribution of several reward models (Figure 1)
- Reward Score distribution comparison before and after training (Figure 2)

We strongly believe that ReMoDetect can be a useful addition to the NeurIPS community, in particular, due to the enhanced manuscript by reviewers’ comments helping us better deliver the effectiveness of our method.

Thank you very much!

Authors.

---

### Decision · Program_Chairs · 2024-09-25

**Decision:**

Accept (poster)

**Comment:**

This work interestingly finds that reward models used in the RLHF stage can detect LLM-generated texts. Based on this, it presents a method that further finetunes the reward model using a challenging dataset rephrased by LLMs. All reviewers liked the work and suggested an acceptance.